# Comparing inter-annual variability in three regional single model initial-condition large ensembles (SMILE) over Europe

Fabian von Trentini[1], Emma E. Aalbers[2,3], Erich M. Fischer[4], Ralf Ludwig[1]

[1]Department of Geography, Ludwig-Maximilians-Universität, Munich, 80333, Germany
[2]Royal Netherlands Meteorological Institute (KNMI), PO Box 201, 3730 AE De Bilt, Netherlands
[3]Institute for Environmental Studies (IVM), Vrije Universiteit, Amsterdam, 1081 HV, Netherlands
[4]Institute for Atmospheric and Climate Science, ETH Zurich, Zurich, 8092, Switzerland

*Correspondence to*: Fabian von Trentini (fabian.trentini@lmu.de)

**Abstract**

For sectors like agriculture, hydrology and ecology, increasing inter-annual variability (IAV) can have larger impacts than changes in the mean state, whereas decreasing IAV in winter implies that the coldest seasons warm more than the mean. IAV is difficult to reliably quantify in single realizations of climate (observations and single model realizations) as they are too short, and represent a combination of external forcing and IAV. Single model initial-condition large ensembles (SMILEs) are powerful tools to overcome this problem, as they provide many realizations of past and future climate and thus a larger sample size to robustly evaluate and quantify changes in IAV. We use three SMILE based regional climate models (CanESM-CRCM, ECEARTH-RACMO and CESM-CCLM), to investigate downscaled changes in IAV of summer and winter temperature and precipitation, the number of heatwaves and the maximum length of dry periods over Europe. An evaluation against the observational data set E-OBS reveals that all models reproduce observational IAV reasonably well, although both under- and overestimation of observational IAV occur in all models in a few cases. We further demonstrate that SMILEs are essential to robustly quantify changes in IAV since some individual realizations show significant IAV changes whereas others do not. Thus, a large sample size, i.e. information from all members of the SMILEs, is needed to robustly quantify the significance of IAV changes. Projected IAV changes in temperature over Europe are in line with existing literature: increasing variability in summer and stable to decreasing variability in winter. Here, we further show that summer and winter precipitation, as well as the two summer extreme-indicators mostly also show these seasonal changes.

## 1 Introduction

In addition to the changes in mean climatological states, the variability of the climate system is an important feature of climate change. This variability of the climate system is subject to various drivers. Variability can be caused by natural forcings on different time scales, such as changes in solar radiation or volcanic eruptions. Variability of single components of the climate system can also be caused by the redistribution of heat and momentum between and within different components (e.g. ocean and atmosphere) of the coupled climate system, referred to as unforced internal variability. Next to these variations, anthropogenic changes in greenhouse gas concentrations contribute to a changing climate. Climate variability can be sampled on different time scales from hours and days up to multi-decadal oscillations.

For impact-analysis of climate change, the future development of inter-annual variability (IAV) is of utmost importance in addition to changes in the mean climate state. Particularly increases in the IAV can be crucial for many impact sectors, as it makes it much harder for stakeholders to plan from year to year. In this study, daily data are used to calculate six indicators: summer and winter mean surface air temperature (tas) and accumulated seasonal precipitation (pr), as well as two indicators for climatological extremes with high societal impact: the number of heatwaves per year (tas-HW-Nr) and the maximum length of dry periods per year (pr-DP-MAX), see Table 1 for definitions. Heat waves can cause an increase of health problems and even fatalities among the population, as well as damages in infrastructure (e.g. highways) and ecological problems, as seen during the most recent heatwaves in Europe (e.g. 2003, 2018, 2019). Long dry periods can have major impacts on ecology, forestry, agriculture, drinking water supply, power plant cooling outages, transport on rivers and many more. All these sectors should implement adaptation strategies to face changing climatic conditions, including IAV.

Early studies with regional climate models from PRUDENCE showed a distinct increase in IAV for the 21[st] century in summer temperatures (Fischer and Schär, 2009; Fischer and Schär, 2010; Vidale et al., 2007) as well as decreasing winter temperature variability (Vidale et al., 2007). Later work with ENSEMBLES models revealed a less pronounced increase in summer temperature variability (Fischer et al., 2012). Analysis of SMILEs also showed future increases in variability of European summer temperatures with increasing global warming (Suarez-Gutierrez et al., 2018; Yettella et al., 2018). Holmes et al. (2016) and Tamarin-Brodsky et al. (2020) also find increasing temperature variability in summer and decreasing variability in winter for the future. European winter temperature variability already today decreased since the pre-industrial era in another large climate model ensemble (Bengtsson and Hodges, 2019).

For large areas of the globe, including Europe, an increase in precipitation variability from daily to multi-decadal time scales is expected due to higher temperatures (Pendergrass et al., 2017). However, Ferguson et al. (2018) find significant changes in the IAV of monthly precipitation only in a small fraction of CMIP5 models for a western European domain until the end of the 21[st] century. Earlier analysis with regional climate models revealed future increases in summer and decreases in winter for IAV of precipitation over similar domains as used in this study (Giorgi et al., 2004).

Uncertainty of future climate projections can stem from at least three sources (Hawkins and Sutton, 2009): emission scenario, model response to a selected forcing and internal variability of the climate system. Internal variability is often referred to as "irreducible

uncertainty" at time scales beyond seasons to decades. While scenario and model response uncertainty have been referred to in many climate simulation experiments (CMIP and CORDEX), the internal variability component had received less attention for many years. In recent years, a new tool for the assessment of internal variability has become quite popular: single model initial-condition large ensembles (SMILEs), where the same model is forced with the same emission scenario several times – with the runs (members) just differing in their initial conditions. This setup is able to isolate the internal variability component from the scenario

and model response uncertainty for the respective model. Based on SMILEs, it has been shown that the contribution of internal variability to total uncertainty of multi-model ensembles (CMIP, CORDEX) can be large, especially for mid-term projections and precipitation (Kumar and Ganguly, 2018; von Trentini et al., 2019) on the regional level.

The terms large ensemble (LE) and SMILE are usually describing the same, but we prefer SMILE as it incorporates the type of large ensemble, which is built upon different initial conditions. Up to now, a number of large ensembles have been produced. Deser

et al. (2020) give the latest overview of the different SMILEs available. However, most studies only use one SMILE for their analysis and the rare comparisons are usually just between two ensembles: similar patterns of internal variability of temperature and precipitation trends for the middle of the 21$^{st}$ century were found for a CCSM3 and an ECHAM5 ensemble over North America by Deser et al. (2014). Martel et al. (2018) showed a consensus of the IAV of annual mean and extreme precipitation in a CanESM2 large ensemble (which is also used for boundary conditions of the CRCM5 in this study, see Data section) and CESM-LE with two

global observational data sets.

All these simulations were performed with global climate models (GCM), and only a few were dynamically downscaled with regional climate models (RCM). Here, we compare three dynamically downscaled large ensembles (all RCP8.5) for Europe. It is the first time that regional large ensembles are compared with respect to forced changes and their internal variability. The added value of RCM simulations is well documented for EURO-CORDEX (Giorgi et al., 2009; Torma et al., 2015; Sørland et al., 2018;

Giorgi, 2019). Downscaled climate data is also a necessity for impact modelling at regional to local scales (e.g. for hydrology, agriculture, biodiversity research) due its more accurate representation of topography, complex coastlines and heterogeneity of land surface properties.

Terminology in the context of climate variability is not always clear in the literature, as the terms natural variability, internal variability, inter-annual variability (IAV) and inter-member variability (IMV) are often used synonymously or mixed up. Here, the

term internal variability describes the variability at timescales from seconds up to multiple decades caused by unforced internal effects of a model or the real world due to the chaotic nature of the climate system only, without incorporating naturally forced variability due to volcanic eruptions and solar forcing. Anthropogenic changes in greenhouse gas and aerosol concentrations as well as natural solar and volcanic forcing can cause changes in the mean climate state that are superimposed by internal variability. Moreover, higher greenhouse gas concentrations can cause changes in the internal variability itself in the future as well – adding

another component to climate change effects.

In this study, IAV is calculated as the standard deviation of anomalies of each member from the ensemble mean (EM), which represents an estimate of the forced response of the respective model. SMILEs have the advantage that the EM is a much better estimate of the forced response than detrending single members. After removing the forced response, the residual IAV equals the

total unforced internal variability – including low frequency variations. We will show that IAV can be well estimated by the IMV
of a SMILE in many cases, as both metrics sample the unforced internal variability of a SMILE, just on different dimensions: IAV
is sampled on the time-dimension of a single member, while IMV is sampled on the member-dimension for each year (see section
3.4). Both IAV and IMV are terms used to describe the more general term 'internal variability' throughout this manuscript. Also,
note that the EM of each SMILE can be considered as the change signal with the highest probability, but which specific member
would become realized depends on internal variability.

The usage of three RCM-SMILEs has some advantages compared to multi-model ensembles consisting of single realizations that
enable us to go beyond the current literature on IAV changes. First, we can better evaluate the models against observations as a) the
forced response of the model is better estimated by the EM and b) we thus reduce the problem of having only one realization of
climate to the observational data side of the evaluation. Second, we can more reliably quantify changes in the IAV, and rule out that
potential changes only occur as a result of internal variability. Additionally, we can better demonstrate when changes are
significantly different from historical conditions. In recent literature, often no significance test of detected changes in inter-annual
variability is performed (e.g. Bengtsson and Hodges, 2019). Many studies just inform about the robustness of change (e.g. by
stippling in maps), measured by the accordance in the sign of change of (usually) 67% of the models of multi-model ensembles (e.g.
Holmes et al., 2016). This does, however, not allow information about the significance compared to a reference climate. Third,
SMILEs allow a better separation of models as they are not only described by one member each. Additionally, we combine these
general SMILE advantages with the higher resolution of RCMs.

The remaining manuscript is structured as follows: First, the model ensembles and the observational data set E-OBS are briefly
presented. Then, the change in mean temperature and precipitation together with the inter-member spread of projected changes is
analysed for each ensemble, as the mean changes are important baseline information for variability changes. Next, the IAV of the
three regional large ensembles is evaluated against E-OBS to assess the abilities of the models to represent observed IAV for the
selected indicators. Finally, IAV in historical climate and future changes in IAV are compared between the SMILEs. This includes
a discussion on different methods to estimate IAV and detect significant changes of IAV. In the main text, most results will only be
presented for Mid-Europe (ME), with references to the other regions and their figures in the Supplementary Material.

## 2 Data

The climate model ensembles each consist of a GCM single model initial-condition large ensemble, which has been dynamically
downscaled over Europe with a single regional climate model: a 50-member CanESM2-CRCM5 ensemble (Kirchmeier-Young et
al., 2017; Leduc et al., 2019), a 21-member CESM-CCLM ensemble (Fischer et al., 2013; Addor and Fischer, 2015; Brönnimann
et al., 2018) and a 16-member EC-EARTH-RACMO ensemble (Aalbers et al., 2018), all forced with the RCP8.5 scenario, resolved
on different spatial resolutions (Table 2). Hereafter we indicate the GCM-RCM combinations with the RCM names only (CRCM,
RACMO, and CCLM). This setup with a shared scenario, but different models, enables us to analyze differences in internal
variability between the three ensembles. Differences in variability may stem from the differences in the resolution of both GCMs

and RCMs, the different domain sizes, the different models, differences in aerosol forcing in the RCM simulations (constant in CCLM and CRCM, transient in RACMO) and in the application of an ocean slab model in the EC-EARTH-RACMO ensemble. RACMO also uses slightly different grid specifications. The domain size of CCLM equals the EURO-CORDEX domain, while CRCM uses a slightly smaller domain, and RACMO only captures central and northeastern Europe (Figure 1). The initialization is carried out differently in the three driving GCM-SMILEs: CanESM2 builds on a hybrid approach, where five members with different ocean conditions starting in 1850 were divided into ten members each using atmospheric perturbations during the initialization in 1950 (see Leduc et al., 2019 for details). The CESM members stem from small atmospheric perturbations of the order of $10^{-13}$ on January 1$^{st}$ 1950 (Fischer et al., 2013). EC-EARTH uses the first 16 days in the year 1850 of an initial run to start the 16 members (Aalbers et al., 2018). These climate model data sets will be compared, but will also be compared to observations: the gridded observational data set E-OBS has daily precipitation and temperature available for Europe (version v12.0, spatial resolution of 0.22° on a rotated pole grid). We use the E-OBS data set because of its availability for Europe and it has similar spatial resolution as the regional climate models under consideration. We accept the known weaknesses of the data set (mostly caused by inhomogeneities in the sparse station network; E-OBS is also known for rather low precipitation fields; see Hofstra et al., 2009), and assume that it is nonetheless suitable for the purpose of this study.

## 3 Methods and results

### 3.1 Spatial aggregation

All indicators (Table 1) are calculated on a grid basis for each ensemble. For comparison, the indicators are spatially aggregated to four regions in Europe, for which all three RCM domains overlap (Figure 1): British Isles (BI), France (FR), Mid-Europe (ME) and the Alps (AL). These regions are well known from other European climate model studies (Lenderink, 2010; Lorenz and Jacob, 2010; Kotlarski et al., 2014; von Trentini et al., 2019), and were introduced by Christensen and Christensen (2007). The procedure of calculating the indicators on the grid level and spatially aggregating them afterwards has the advantage that no regridding of data is needed. However, the different spatial resolutions of the models alone can potentially lead to higher variability in the 0.11° data (CRCM and RACMO), compared to the 0.22° (E-OBS) and 0.44° (CCLM) data. This is especially the case for spatially heterogeneous variables and indicators (Giorgi, 2002; Kendon et al., 2008). The indicators in this study, however, have relatively low spatial heterogeneity (seasonal temperature and precipitation, heatwaves and dry periods are rather large-scale phenomena), where the range of spatial resolutions of the data used here (between 0.11° and 0.44°) is not expected to be significantly sensitive. The effect of regridding before the calculation of indicators is shown by a short experimental analysis, where one year of five members of the 0.11° CRCM data is regridded to 0.44° (simply averaging 4x4 grid cells each), before the indicators are calculated. The results show that the effect of regridding on the IMV is indeed minor for the indicators considered (Supplementary Material, Figure S1). The approach of direct regional aggregation of the indicators calculated on the grid level is therefore applied for the further analysis of this study.

**3.2 Ensemble spread of projected mean climate change**

Before analysis of IAV, simple scatter plots of the changes in the mean climatological states of each member for temperature and precipitation for summer and winter between 1980-2009 and 2070-2099 are shown for the Mid-Europe region (ME), see Figure 2 and Figure 3. They give a first impression on the spread of projected changes between the members of the SMILEs and on the differences in the mean changes between models. We test the similarity of means between all models with a two-sample t-test ($\alpha$=0.05) and the similarity of spreads with a Brown-Forsythe Test (BF test with $\alpha$=0.05) on equal variances. The BF test does not show significant differences in variance for both temperature and precipitation in winter and summer for all model combinations. The spread of signals between members of one SMILE can be solely attributed to the internal variability of the respective model. When models generally agree on the spread of members, the confidence in the models' ability to represent internal variability gets higher.

In summer in ME, all models show decreasing precipitation; between -3 % and -16 % for RACMO, and -14 % to -35 % for CRCM and CCLM (Figure 2). Increases in summer temperature between 3 °C and 5 °C are projected by RACMO and CCLM, while CRCM shows much higher changes between 5 °C and more than 6 °C. Thus, RACMO and CCLM show similar changes in temperature (although statistically different in their means), while CCLM and CRCM show similar changes in precipitation. The spread of changes for both temperature and precipitation of RACMO and CRCM are similar, both in terms of standard deviation and total range, while CCLM shows higher standard deviation and total range (Table 3). Similar results as discussed here for Mid-Europe (mean changes and spread) are found for France and the Alps (not shown), with the largest decrease in summer precipitation over France and the strongest warming over the Alps. The British Isles region shows less pronounced changes for both temperature and precipitation (although consistent in sign). CRCM shows closer similarity of precipitation decreases to RACMO in BI rather than to CCLM, as it is the case for the other three regions ME, FR and AL.

In winter, all models project increasing precipitation (1-32 %) and temperature increases between 1.4 °C and 5 °C by the end of the 21[st] century (Figure 3). RACMO and CRCM show similar standard deviation and range of temperature and precipitation changes again, together with similar mean changes as well (significant for temperature, but not for precipitation). CCLM shows distinctly smaller changes in combination with a smaller spread of changes (Table 3). Similar results also appear for FR, AL, and BI, although some members of CCLM and CRCM also project a slight decrease in precipitation in these regions.

**3.3 Evaluation against E-OBS**

For the evaluation of the models' IAV against E-OBS, we apply an approach proposed by Suarez-Gutierrez et al. (2018) and Maher et al. (2019). For the observations and for each model and member separately, the anomalies relative to the reference period 1961-1990 are calculated for the years 1957-2099 and 1957-2015 in E-OBS, respectively (Figure 4). Model mean state biases of the indicators, which can be quite large (see Figure S2) are thereby removed. For each year, we then plot the ensemble median, minimum and maximum member, the area between the 12.5[th] and 87.5[th] percentile, within which 75 % of the members are situated, and the E-OBS data. For a perfect model, the E-OBS data is expected to occur normally distributed within the range spanned by the

ensemble, is concentrated in the inner 75%, several years in between the minimum and maximum of members, but also outside this

range from time to time. If the E-OBS data concentrates too much inside the total range or even the 75% area, the variability of the ensemble overestimates the observational variability. Contrary, if too many E-OBS data points exceed the ensemble spread, the SMILE underestimates observational variability. To quantify this further, the probability density function of the anomalies in the period 1957-2015 are plotted for each member and E-OBS separately. The functions are estimated probability densities based on a normal kernel function, similar to an approach by Lehner et al. (2018).

The forced response (ensemble median) increases for all indicators analysed, except for summer precipitation, which decreases in all models, and no clear change of pr-DP-MAX in RACMO. Note that this approach does not only compare the IAV of the models and E-OBS, but also the forced response in the historical period. Differences in the distributions can thus also arise from a false representation of the forced response in a model, compared to the trend of E-OBS. On the other hand, if the modelled and observed distributions largely coincide, both the forced response and the IAV are well represented by a model. All three models generally

seem to reproduce the forced response in the historical part quite well, as the models are consistent with the trends of the E-OBS points (e.g. increase of tas-JJA). Only for summer precipitation (pr-JJA), all models show a decrease of the forced response whereas E-OBS shows no significant negative trend. However, in all ensembles, not all members show decreasing trends. The observations may thus still be consistent with the simulated forced response.

The comparison of E-OBS and the three SMILEs during the historical period from 1957-2015 in Mid-Europe (ME) shows largely

good representations of IAV in the ensembles, as seen by well distributed E-OBS points within the 75 % range (12.5-87.5% quantile) and minimum and maximum range of the ensembles (Figure 4). However, a too strong clustering of the E-OBS points in the 75 % area occurs for winter precipitation in CRCM (97 % fall inside) and number of heatwaves in CCLM (90 %), meaning the simulated IAV is too high. On the other hand, too many outliers beyond the minimum and maximum members appear in winter temperature in CCLM (22 % outside of total range), winter precipitation in RACMO (17 %) and maximum duration of dry periods in CRCM

(10 %), i.e. for these models and indicators the simulated IAV is too low. To demonstrate this further we calculate probability density functions of the annual anomalies for each member and E-OBS (Figure 5). Note that probability density functions could also be somewhat inflated by the underlying mean trend, but we expect this effect to be small because trends in the observational period are small and largely consistent between models and observations. To evaluate the ability of the SMILEs to represent observational IAV, we test whether the E-OBS distribution looks like a possible member of the respective ensemble. The

observations are not expected to fall near the ensemble median, but rather should be ideally indistinguishable from a random additional member of the ensemble, since E-OBS only represents one possible realization of historical climate. Significant differences can be seen for the already mentioned examples: the distribution of CRCM in winter precipitation is much broader than the E-OBS distribution, whereas the winter temperature distribution for CCLM concentrates too much in the middle compared to E-OBS.

Similar results as in ME can be found in the other three regions as well (Figures S3-S5), with only several cases where the E-OBS distributions show a distinctly different shape than all members of the ensembles (Figure 6 for AL and Figures S6 and S7 for BI and FR, respectively), especially for the maximum duration of dry periods of CCLM in France. This is not too surprising, as the

maximum duration of dry periods is an extremely sensitive indicator, because of its potentially extreme differences in magnitude between (model) years/members (one wet day can make a huge difference). The other two SMILEs are able to represent the E-OBS variability for this indicator in France though. Other remarkable features are the underestimation of variability in RACMO for all six indicators in the British Isles region (Figure S6), as well as the relatively good performance of the models for the Alps (Figure 6), which is probably the most difficult region for a model to represent correctly due to the strong spatial heterogeneity. However, the Alps show some "outlier-members" with distinctly different distributions in the ensembles (e.g. winter precipitation in CRCM), which cannot be found in the other regions – at least not this pronounced. These outliers demonstrate how large the influence of internal variability between members can be in a single realization of climate, as these outlier members just deviate from all other members by their initial conditions. Estimating the IAV of a model thus needs a large number of members, as even with 49 members that give a uniform range of distributions (pr-DJF in CRCM5 in the Alps, Figure 6), one single additional member can change the picture and add more information on the range of IAV for the respective model. The evaluation of E-OBS gets rather difficult in these cases, as the methodology is based on the assumption that the E-OBS distribution should somehow "fit" to the uniform range of distributions of the model. If the E-OBS data would show such an outlier behaviour, it means that the one realization of climate variability as seen by the E-OBS data might still be part of a SMILE's range of possible variability manifestations. However, from a probability perspective, the conclusion of similar variabilities gets rather unlikely. It just makes it harder to prove that E-OBS has a different distribution than all members of a SMILE.

### 3.4 Projected changes in internal variability and the connection between IAV and IMV

The temporal development of the internal variability is important information along with the underlying forced response (change in the EM) for a better understanding of changing climatic conditions. We discuss three possible ways to describe changes in the internal variability on annual timescales within a SMILE. All three methods are based on the application of a Brown-Forsythe test (BF test) on equal variances. While IAV and IMV are expressed as standard deviations (std), the BF test analyses differences in the variance, which is just the square of std. In the cases of IAV (methods 1 and 3), moving time periods of 30-year length, shifted by one year each (1961-1990, 1962-1991, …, 2070-2099) are used. For the second method, IMV is sampled over the dimension of the ensemble size per year. Thereby we test if the internal variability changes significantly over time. Differences in the methods arise from the different data samples used for the testing.

The first method is based on the methodology that one would choose for single members and observations. By looking at the IAV for different periods within one member, changes in IAV can be detected. Usually the forced response is taken out of the data by fitting a polynomial to the data and only using the residuals. However, the estimate of the forced response of a model based on only one member may deviate from the true forced response (Lehner et al., 2020). Therefore, we choose the EM as an estimate of the forced response and use the residuals from each member with respect to the EM for the BF test. The BF test results is a Boolean information for each member and each moving period on whether the variance has significantly changed with respect to a reference period (here: 1961-1990) or not. This information can be used to show the percentage of members with a significant change (separated for positive and negative changes) in each period. The advantage of SMILEs within this approach is the better estimate

of the forced response and a more robust detection of changes, as they are built on multiple members. One member alone could be an outlier in its representation of (changes in) IAV just as it could be for the trend. The method is sensitive to the chosen reference period of course, as the variance of this period determines the baseline variance. Since we use moving periods, the results do not change significantly when using different periods starting in the 1960s.

For the second method, we make use of the assumption that the IMV for a given year is a good approximation for the IAV in a period around that year (e.g. ± 15 years to get a sample size of the typical 30 years for climate analysis). The sampling of variability is thus not based on consecutive time series within each member, but on a compound of annual data for one year from all members of a SMILE. The IMV is also based on residuals from the EM as for IAV. Under the assumption of small influence of low frequency variability, IMV should be a good estimate of total unforced IAV, as both sample the annual variability during a similar state of

climate for a given time horizon. This concept is particularly relevant in the presence of non-linear forcing. For instance, the response to a volcanic eruption cannot be separated easily from unforced IAV. In addition, the anthropogenic forcing since 1950 has not been linear in time. Using IMV is an elegant way to get around this challenge. Some recent publications support the concept of using IMV as an approximation of IAV, although the two have different background meanings: while IAV has a physical meaning and represents the variability of a consecutive sequence of weather phenomena, IMV is a measure of variability without a direct physical

meaning (Nikiéma et al., 2018).

In Leduc et al. (2019) the authors state that "In the case of a climate system under transient forcing, the use of [IMV equation] to assess temporal variability using the inter-member spread involves weaker assumptions than calculating the residual temporal variability from detrended time series." (Leduc et al., 2019, p. 681), based on the study by Nikiéma et al. (2018). A recent publication by Wang et al. (2019) even concludes that the IMV of winter sea level pressure over Eurasia in a SMILE is driven by the same

mechanisms as observed IAV via an EOF analysis. Another example is the analysis of seasonal mean and heavy precipitation in Europe where long-term variations are small compared to the IAV in the RACMO ensemble (Aalbers et al., 2018). A comparison of IAV and IMV in each ensemble is carried out by comparing the means and standard deviations of these two variability metrics – calculated over different dimensions of the ensemble data. The IAV is calculated for each member during a 30-year reference period (1980-2009) and three future periods. The mean and standard deviation of these 50/21/16 values is calculated for IAV. The IMV is

calculated for each of the 30 years of the respective period between the 50/21/16 members, leading to a mean and standard deviation, calculated from these 30 values. The mean and standard deviation of IAV and IMV are indeed very similar for all indicators, periods and regions (exemplary shown for winter temperature in Figure 7). Especially the similarity in future changes suggests a similar response to external forcing for the two variability metrics IAV and IMV. The IMV has the advantage that it is insensitive to inflation effects of the variability due to an existing trend and forced effects like cooling after volcanic eruptions for example. However,

although IAV and IMV seem to be similar in many cases (see also literature above), they can potentially also differ under special circumstances in the external forcing like volcanic eruptions. Note that according to our results IMV is always slightly larger than IAV. This may be caused by two factors. First, detrending the time series is more likely to remove than to add some of the variability, and affects only IAV. Second, also without detrending the data, in the presence of low-frequency variability, IAV is likely smaller

than IMV, which has no auto-correlation in the underlying data. For the variables considered here, differences are small though,
implying that the low-frequency variability is indeed small compared to the high-frequency variability.

Given the similarity of IAV and IMV, the third approach pools together the annual anomalies from the EM from all members for a given 30-year period (30 times the ensemble size, e.g. 30x21 values for CCLM). It is therefore a mixture of IAV and IMV, enabling a more robust BF test result for changes in variance by a larger sample size.

While the interpretation of temperature based indicators is always based on absolute anomalies from the EM, it can be useful to look at both absolute and relative anomalies from the EM for precipitation based indicators (in contrast to the previous evaluation against E-OBS, where they were only absolute anomalies). Relative anomalies thus give information on how much the standard deviation changes with respect to changes in the EM. For example, a stable IMV in absolute terms will result in a decrease of the relative IMV when the EM increases. Increasing relative IMV, together with an increasing EM on the other hand means that the internal variability is increasing even more than the mean.

The percentage of members with significant changes in IAV as function of time is shown in Figure 8 for all indicators, for Mid-Europe. Significantly decreasing IAV for winter temperature and increases for summer temperature and heatwaves are found, but only for a minority ($< 50\%$) of the members for all models, even at the end of the 21$^{st}$ century. While all three models point to the same direction of change, percentages differ substantially. For winter and summer precipitation, an even smaller percentage of members shows significant changes in IAV, and there is no clear direction of change in any model. Only CRCM for pr-JJA shows an increasing number of members with significant positive changes throughout the second half of the 21$^{st}$ century. For dry periods, RACMO has a very small number of members showing significant changes in both directions, while CRCM and CCLM show marked increases in the number of members with significant positive changes in IAV throughout the 21$^{st}$ century. For the last period 2070-2099, even all members of CCLM show significant increases.

The temporal evolution of IMV (relative to the EM for precipitation-based indicators, Figure 9) generally supports the direction of changes as seen by the method using the percentage of members with significant changes in IAV. However, when testing for significant changes in the variance between members, hardly any of the changes are significant. CRCM shows significant changes in the majority of years for tas-DJF from 2040 on, for tas-JJA from 2080 on and for pr-JJA from 2060 on. As the IMV is calculated for each year, the plot shows the noise in the IMV per year, which can be large.

Figure 10 shows the change in variability determined from the pooled annual anomalies from the EM for moving 30-year periods from all members. This means, all 30 anomalies from all members are pooled together before calculating the standard deviation (i. e. pooled IAV) and tested for significant changes in the variance with a Brown-Forsythe Test. Given the much larger sample size per 30-year period, in contrast to the two former methods, we can now see significant changes in many combinations of indicator and model (Figure 10). As expected from the previous two methods, internal variability decreases for winter temperature and increases for summer temperature and the number of heatwaves. In contrast to the former methods, however, significant changes can be detected earlier. In these cases, the internal variability has already changed significantly in the historical simulations of the SMILEs or it changes in the present/near future around 2020. The internal variability in the number of heatwaves increases until about 2010-2030, reaches a plateau for about 30-40 years and then decreases again. This behaviour can be explained by the forced

response of the indicator, which shows strong increases until around 2060, when the number of heatwaves stabilizes around 6 (and even decreases afterwards in CRCM, Figure S2), because the heatwaves get so long that their number per year cannot increase

anymore. This is especially true for CRCM, where the mean duration of heatwaves at the end of the 21st century is much longer than for CCLM and RACMO and about 16 days (not shown), leading to a rough estimate of 6*16=96 heatwave-days per year, equal to about three months. Since heatwaves are defined by the 95th percentile of temperature in the reference period (thus describing extreme conditions), the former extreme heat becomes a regular condition during the summer months at the end of the 21st century in CRCM. For the pooled IAV, both absolute and relative changes of IAV are shown for precipitation-based indicators to

demonstrate the effect of the two different approaches. For pr-DJF, CRCM does not show any change in absolute IAV, while this stable behaviour in combination with the increase of pr-DJF in the EM leads to a decreasing relative IAV, which is significant from the early 21st century onwards. CCLM and RACMO show increasing variability in absolute terms, but changes are significant for RACMO only, from ~2060 onwards. For both CCLM and RACMO there is no clear change in IAV relative to the change in EM for pr-DJF. Note that while RACMO shows the lowest absolute IAV it shows the highest relative IAV. This originates from the

lower EM for winter precipitation in RACMO compared to CRCM and CCLM, which both have quite distinct wet biases (Figure S2). For summer precipitation, absolute IAV increases according to all models, while EM decreases. Changes in absolute IAV are largest and significant for CRCM and RACMO from ~2000, respectively ~2045 onwards. For CCLM changes are not significant. Owing to the decreasing EM, increases in relative IAV are significant for all models and significant changes occur earlier in time (~1970 for CCLM, ~1990 for CRCM and ~2040 for RACMO). The changes in EM and IAV in both summer and winter have also

been detected by Pendergrass et al. (2017) for CMIP5 and CESM-LE precipitation data in extratropical regions. IAV of pr-DP-MAX does not change according to RACMO, while CRCM and CCLM show distinct increases that go hand in hand with increases in the EM that is also much stronger in these two models than in RACMO (Figure 4). The changes are significant for relative IAV later in time than for absolute IAV.

The above mentioned results are mostly valid for the other regions as well. Differences in the magnitudes of variability and its

changes are shortly discussed in the following (see Figures S8-S10): ME shows higher winter temperature variability than the other three regions, especially than BI. Lower levels of variability compared to the other regions occur over the British Isles for winter temperature and winter precipitation (relative to EM). The Alps show a smaller variability for the number of heatwaves than the other regions. The variability of pr-DP-MAX for all three ensembles is similar to ME in AL, while BI and FR hardly show any significant changes. If all regions are considered, RACMO generally has the highest internal variability in winter and the lowest

variability in summer for temperature and precipitation (relative to EM), while CCLM has the highest internal variability for summer temperature and precipitation (relative to EM) as well as for heatwaves and dry periods (both absolute and relative to EM). Significant changes generally occur similar to ME for winter and summer temperature and precipitation (both absolute and relative to EM). Changes in the number of heatwaves are not significant in CCLM in all three regions and in RACMO in AL.

The first method, testing the percentage of members with significant changes in IAV, gives a good overview on the behaviour of

the members in general. It can, however, just inform about the direction of change in IAV. Additional information on the magnitude of the changes in IAV are needed to get a whole picture. The second method using IMV is in general agreement with the first method

when looking at the direction of change. It incorporates the magnitude of internal variability and can show significant changes in the IMV in the same figure. The variations of IMV from year to year are relatively large. Therefore, also the BF test results largely depend on the choice of a reference year to test all other years against. Changes from one year to the next can give very different

results. Unstable significance testing is the result. Method 1 is less sensitive to the reference given the overlapping periods. Considering the sample sizes in this study, the best results can be obtained with method 3, where sensitivity can be tested with a much larger sample size. For method 1 it is 30 values per period, for method 2 it is 16/21/50 values per year, and for method 3 it is the product of both 30 years and the member size of the respective ensemble. However, if the sample size would be larger for the first two methods, they will probably also result in the detection of significant changes, e.g. when using more members.

To see how the ensemble size impacts the results for the pooled IAV, we reduce the largest ensemble (CRCM with 50 members) to the size of the other ensembles (16 and 21) and repeat the analysis for ME. Even for only 16 members, the changes in CRCM are still significant, where they are for 50 members. However, the detection of significant changes is only possible at a later time horizon (Figure S11). Tests with a number of ensemble sizes, suggest that around 10 members are sufficient to detect the significance of changes, and around 20 to detect the timing of these significant changes additionally.

**5 Discussion**

The number of SMILEs available for the quantification of internal variability in this study is still relatively small – we only used three GCM-RCM combinations (to the knowledge of the authors these three ensembles are the only regionally downscaled SMILEs over Europe). More simulations with RCM-SMILEs could help to make results even more robust – especially for winter precipitation and dry periods, where the three ensembles do not agree on the change in variability.

The effect of regional aggregation after the calculation of indicators on the grid level, and the potential effects of the original resolution of different data sets on the internal variability seem to be minor for the selected indicators, as seen in the experimental analysis conducted on a subset of the data (Figure S1). This estimate of sensitivity to differing spatial resolutions might be conservative, however. Nevertheless, the methodology seems to be suitable for the selected indicators of this study.

Methods based on anomalies from the EM are chosen in order to compare results despite different biases in historical and future

mean climate states. It can, however, not be ruled out that differences in the variability may originate from mean state biases of the models. CRCM for example shows much higher precipitation sums than the other two ensembles, leading to higher variability in absolute terms. The normalization with the ensemble mean is covering these differences in absolute amounts. In the end it largely depends on the definition of variability: is one interested in absolute deviations [mm] or in the fluctuations in relative terms [%]? Results are sensitive to a relative versus an absolute definition or vice versa. The relative approach has the advantage that it allows

for a fair comparison of models with different mean precipitation amounts. This is also why a recent publication by Giorgi et al. (2019) gave preference to the relative definition, for example.

The scatter plots of projected changes for seasonal temperature and precipitation (Figure 2 and Figure 3) show both agreement and dissent, but usually at least two of the three models show similar ranges for one variable. Better agreement might be possible when

comparing the data sets not for a fixed period, but for periods with the same global warming level in each driving GCM.

We find that the large ensembles analysed here generally represent observed IAV correctly, but care needs to be taken during the analysis for specific regions and indicators. Both cases of many individual members showing higher and lower variability compared to observational IAV can be found for all ensembles for specific indicators and regions. However, the single observed realization of historical climate makes it difficult to evaluate systematic errors of the ensembles, as the E-OBS distribution is not necessarily

representative for the perfectly-sampled IAV. It would be interesting to compare large ensembles against an observational large ensemble as proposed by McKinnon and Deser (2018) to better see systematic deficiencies of large ensembles compared to observations.

The results found for changes in IAV are generally in line with existing literature over Europe. We likewise find increasing variability for the summer indicators tas-JJA, pr-JJA (Fischer and Schär, 2009; Fischer and Schär, 2010; Vidale et al., 2007; Yettella

et al., 2018; Suarez-Gutierrez et al., 2018) and decreasing variability for the winter indicators tas-DJF and pr-DJF  (Bengtsson and Hodges, 2019; Holmes et al., 2016). The summer extreme indicators tas-HW-Nr and pr-DP-MAX also show increased variability in two of the three models, in conjunction with increases in their mean states. Several mechanisms contribute to the changes in all indicators. For changes in the summer temperature IAV, land-atmosphere coupling is becoming more important in central/northern Europe in the future, because the transitional zone between dry and wet climates moves northwards from the Mediterranean region,

leading to enhanced alternation of dry and wet summer soil moisture (Seneviratne et al., 2006; Fischer et al., 2011). Moreover, stronger warming over land than over the oceans causes the land-ocean temperature gradient in summer to increase. This results in increased variability in thermal advection, which is suggested to play a role in the increase in temperature variability in Europe as well (Holmes et al., 2016). Analysis of observations shows that in the Mediterranean more than half of summer temperature variability can be explained by large-scale atmospheric circulations and sea surface temperatures (Xoplaki et al., 2003). The decrease

in winter temperature IAV is suggested to be influenced by changing circulation patterns (Vautard and Yiou, 2009), and a decrease in variability of advected heat due to the decrease in the winter land-ocean temperature gradient (Holmes et al., 2016) and arctic amplification and sea ice loss (Screen, 2014; Sun et al., 2015; Tamarin-Brodsky et al., 2020), even under unchanged circulation variability. (Holmes et al., 2016; Tamarin-Brodsky et al., 2020).

The increase in summer precipitation variability that would not be expected under decreasing mean summer precipitation, might be

caused by a reduction in the number of wet days (>1mm) that exists in all three ensembles (not shown), as discussed by Räisänen (2002). A reduction of wet days implies an increase in variability since the seasonal precipitation sum gets more dependent on individual precipitation events.

Land-atmosphere feedback mechanisms are not yet fully understood, and there are still improvements needed in their implementation in earth system models and regional climate models (Vogel et al., 2018). Uncertainties in the future regional

development of heatwaves and dry periods are thus rather large (Miralles et al., 2019). Nevertheless, increasing frequency, intensity and variability in the number of heat waves as projected by the SMILEs using RCP8.5 in this study seem plausible, although the

magnitudes can be uncertain. The strong increase in the maximum length of dry periods in two of the models is not necessarily what could be expected. While the length of severe dry periods increases in the future for southern Europe, central and northern Europe do not show any change in the EURO-CORDEX data for dry period length (Jacob et al., 2014). Analysis of precipitation changes shows that both CRCM and CCLM (the RCM itself, not the SMILE) are on the dry end of projections for summer precipitation (von Trentini et al., 2019). This might be related to sensitive implementations of land surface modules in these two RCMs. RACMO does not show an increase in the maximum length of dry periods.

Although beyond the scope of this paper, which only analysed the manifestations of internal model variability in surface variables (tas, pr and associated indicators), there is a need for a better understanding of the mechanisms leading to the model-inherent characteristics of internal variability, and why differences between the models appear (e.g. circulation patterns, ocean characteristics).

Using SMILEs for studying changes in IAV allows for much more confident statements on the direction, magnitude and emergence of changes, when using a certain model and RCP scenario. Analysing the individual members, significant changes in IAV are found in less than half of the members for almost all indicators and ensembles, even at the end of the 21$^{st}$ century (Figure 8). However, pooling the data of all ensemble members, all three ensembles show significant changes in internal variability of most indicators, and often from early in the 21$^{st}$ century onwards (Figure 10).

Thompson et al. (2015) showed that a statistical model based on a historical period could be as good as a SMILE for predicting future variability of seasonal temperature and precipitation trends up to 2060. The detection of significant increases of internal variability in summer and winter temperature much earlier than 2060 (Figure 10) challenges these assumptions. For precipitation, however (especially absolute changes), the changes are often not significant before 2060, confirming the results of Thompson et al. (2015).

## 6 Conclusions

There is an increasing interest of the scientific community to use single model initial-condition large ensembles in a wide variety of applications, ranging from deeper levels of understanding of natural climate variability to impact assessments in different fields. The rich data basis which these ensembles provide for the analysis of internal variability is very valuable and enables new insights into this critical part of the climate system. Especially future changes can be better set into context. The effects of dynamical downscaling of GCM large ensembles with regional climate models are not yet sufficiently explored. Further research is needed in this direction to see whether and by how much the internal variability is altered in the RCM simulations of a respective GCM large ensemble. However, downscaling is an important step to make climate simulation information attractive for local adaptation research and impact modellers. The results from this study can be helpful for these research communities to better understand and quantify the role of IAV in the climate system. Especially increases in variability as seen for summer temperature, relative summer precipitation, heatwaves and dry periods in most regions and models, can be a huge burden for sectors like agriculture, ecology and hydrology.

The evaluation and comparison of the three RCM-SMILEs in this study gives a first overview on the agreement of the SMILEs with observations and among each other. The moderate agreements in both cases suggest that the internal variabilities of RCM-SMILEs at the regional scale are good approximations of the IAV of the climate system in Europe. The direction of changes in internal variability is also mostly the same between the ensembles, suggesting a relatively robust signal. While the "summer indicators" mostly show increasing variability in the future, winter temperature and precipitation show decreasing variability or no change. The change in variability is potentially impact-relevant as it suggests that the most extreme summers and winters may warm stronger than the corresponding mean.

Despite an increasing number of studies that compare SMILEs (Deser et al., 2014; Martel et al., 2018; Rondeau-Genesse and Braun, 2019; Deser et al., 2020; Lehner et al., 2020), one limitation of many publications using SMILEs is the use of only one model, and thereby one estimate of internal variability, leaving it unclear how representative the results are. Although the respective SMILE is usually evaluated against observations in these studies, the uncertainty in future changes of IAV cannot be quantified in the same way as in this study. A further challenge is also the fact that low frequency variability at decadal and multi-decadal time scales remains uncertain and cannot be rigorously evaluated against observations due to the relatively short observational record and the difficulty of separating forced changes from unforced internal variability in observations.

So, do we need to go the way CMIP and CORDEX went and apply a multi-model-multi-member ensemble in all studies? First approaches in this direction are currently ongoing for the GCM SMILEs (e.g. Lehner et al., 2020). This would certainly be an ideal way to better understand and quantify sources of uncertainties, but is challenging for two reasons: a) the computational resources to perform these experiments on regional scale are limited and the CORDEX matrix of scenario, GCM and RCM could already not be filled for this same reason, and b) the use of hundreds of simulations would also be very challenging for analysis and impact modellers.

Overall our results underline the great potential of SMILEs in quantifying the changes in IAV and when they become significant, also at the regional scale.

**Table 1: Indicators and their definitions**

| Indicator | Used variable | Definition |
|---|---|---|
| tas-JJA | tas | Summer mean temperature (June-August) |
| tas-DJF | tas | Winter mean temperature (December-February) |
| tas-HW-Nr | tas | Number of heatwaves per year; a heatwave is defined as a minimum of three days above the 95$^{th}$ percentile of daily mean temperature of the reference period; no filtering on summer months is applied at any stage, however, by definition the heatwaves will occur during summer in the reference period. They might, however, extent to spring and fall under RCP8.5 |
| pr-JJA | pr | Summer precipitation sum (June-August) |
| pr-DJF | pr | Winter precipitation sum (December-February) |
| pr-DP-MAX | pr | Maximum length of a dry period per year; dry periods are a minimum of 11 consecutive days with every day showing less than 1 mm of precipitation; no filtering on summer months is applied at any stage, the periods can thus also occur in winter (but this is rather unlikely in Europe) |

**Table 2: Specifications of the three ensembles used in this study**

| | 'CRCM' | 'CCLM' | 'RACMO' |
|---|---|---|---|
| Scenario | RCP8.5 | RCP8.5 | RCP8.5 |
| GCM | CanESM2 | CESM 1.0.4 | EC-EARTH 2.3 |
| GCM Resolution | 2.8° | 2.0° | 1.0° |
| RCM | CRCM5 | CCLM4-18-7 | RACMO22E |
| RCM Resolution | 0.11° | 0.44° | 0.11° |
| No. of members | 50 | 21 | 16 |

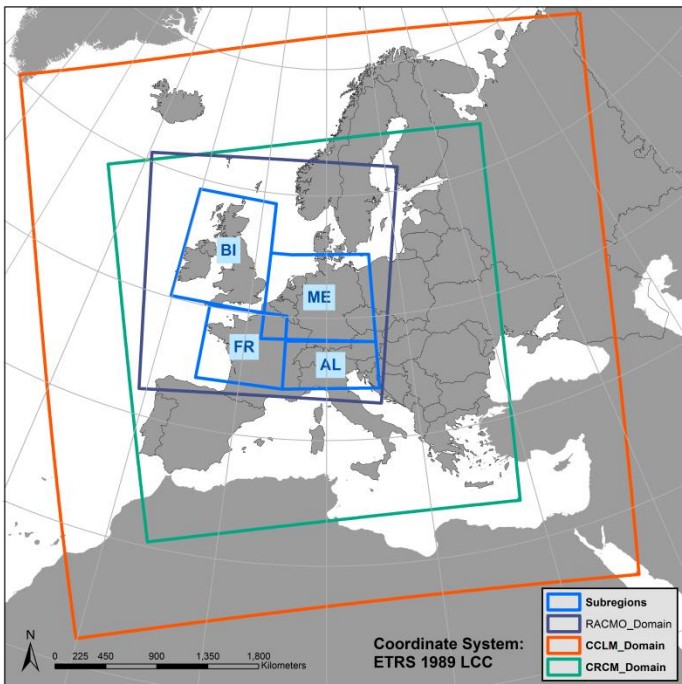

**Figure 1: Domains of the three RCMs and the boundaries of the four analysis regions; BI=British Isles, FR=France, ME=Mid-Europe, AL=Alps; the CCLM domain matches the EURO-CORDEX domain**

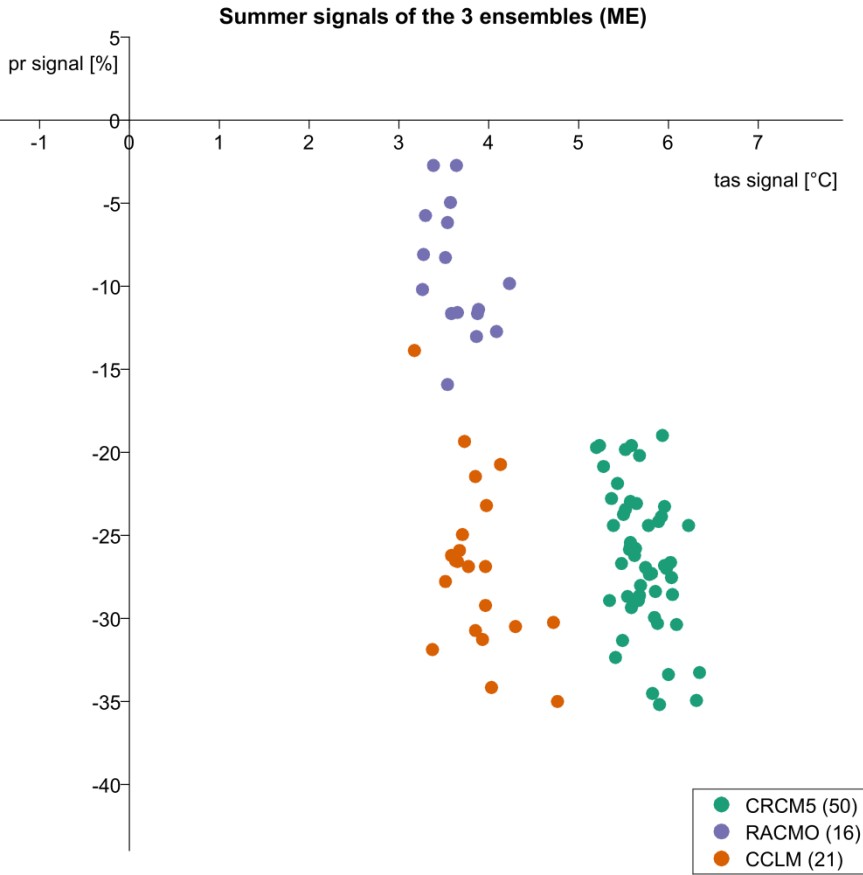

**Figure 2: Change in mean summer temperature and precipitation for every member of the three ensembles in Mid-Europe (2070-2099 against 1980-2009). Changes are relative to each members' value in 1980-2009 for precipitation, while temperature changes are absolute**

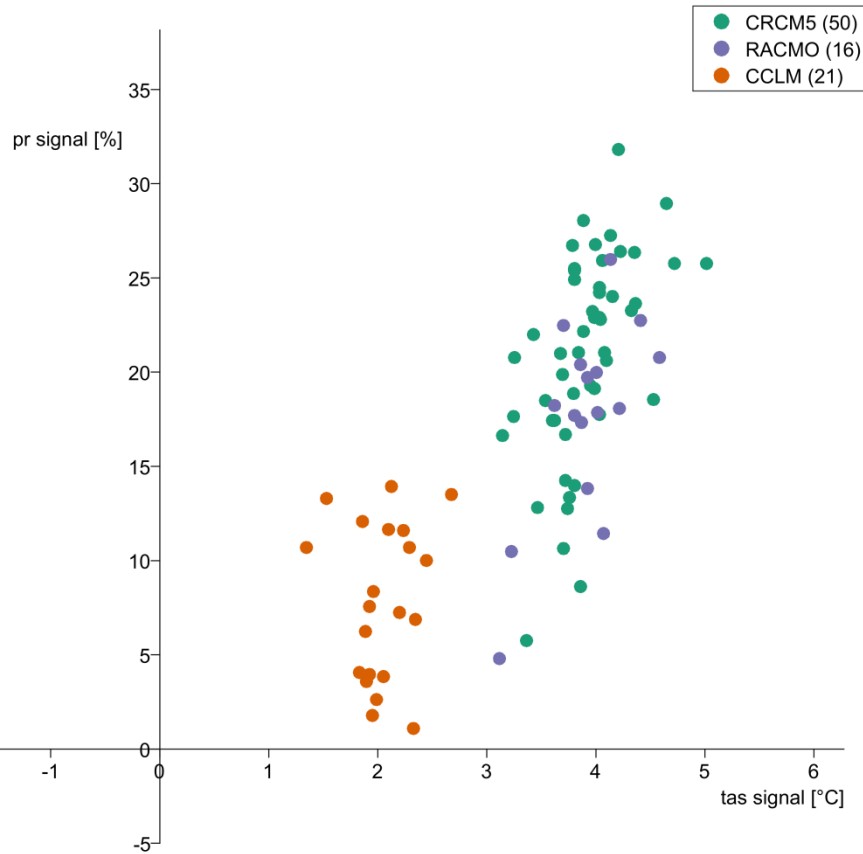


**Figure 3: Change in winter temperature and precipitation for every member of the three ensembles in Mid-Europe (2070-2099 against 1980-2009). Changes are relative to each members' value in 1980-2009 for precipitation, while temperature changes are absolute**

**Table 3: Standard deviation and total range for changes in Figure 2 and Figure 3**

| | tas [°C] | | | pr [%] | | |
|---|---|---|---|---|---|---|
| | CRCM | RACMO | CCLM | CRCM | RACMO | CCLM |
| summer std | 0.27 | 0.28 | 0.39 | 4.2 | 3.8 | 5.1 |
| summer range | 1.16 | 0.96 | 1.59 | 16.2 | 13.2 | 21.1 |
| winter std | 0.37 | 0.38 | 0.30 | 5.5 | 5.3 | 4.2 |
| winter range | 1.87 | 1.47 | 1.33 | 26.1 | 21.2 | 12.9 |


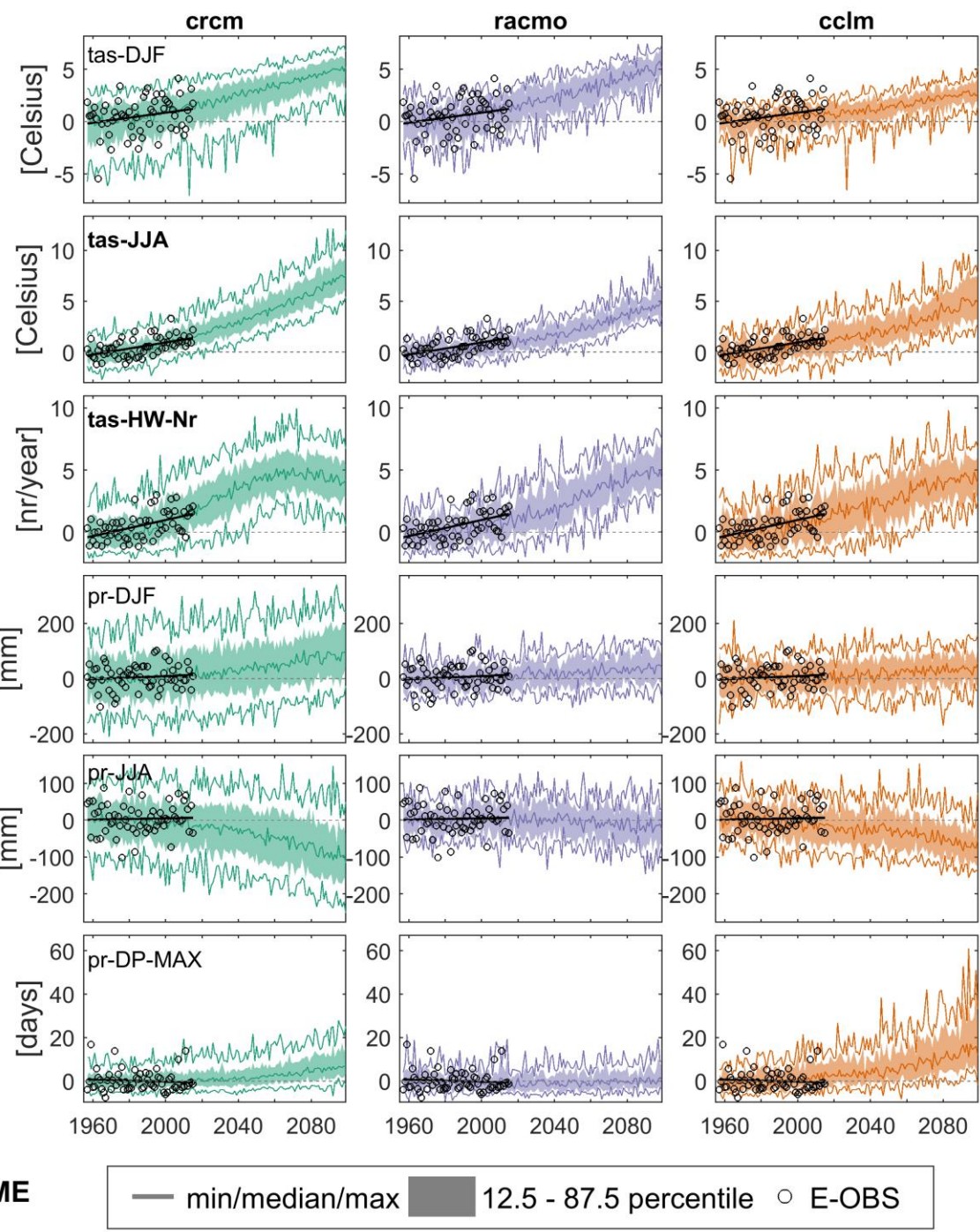

**Figure 4: Anomalies from 1961-1990 of the 6 indicators in Mid-Europe (ME) for E-OBS (circles 1957-2015) and the three ensembles (1957-2099), represented by the median, minimum and maximum (solid lines) of the ensemble and an area from the 12.5th and 87.5th percentile,**

 spanning the range of the inner 75 % of the members (shading). Black lines show the linear trend for the E-OBS points. The indicator names are in bold when the trend is significant using a Mann-Kendall test (alpha=0.05).

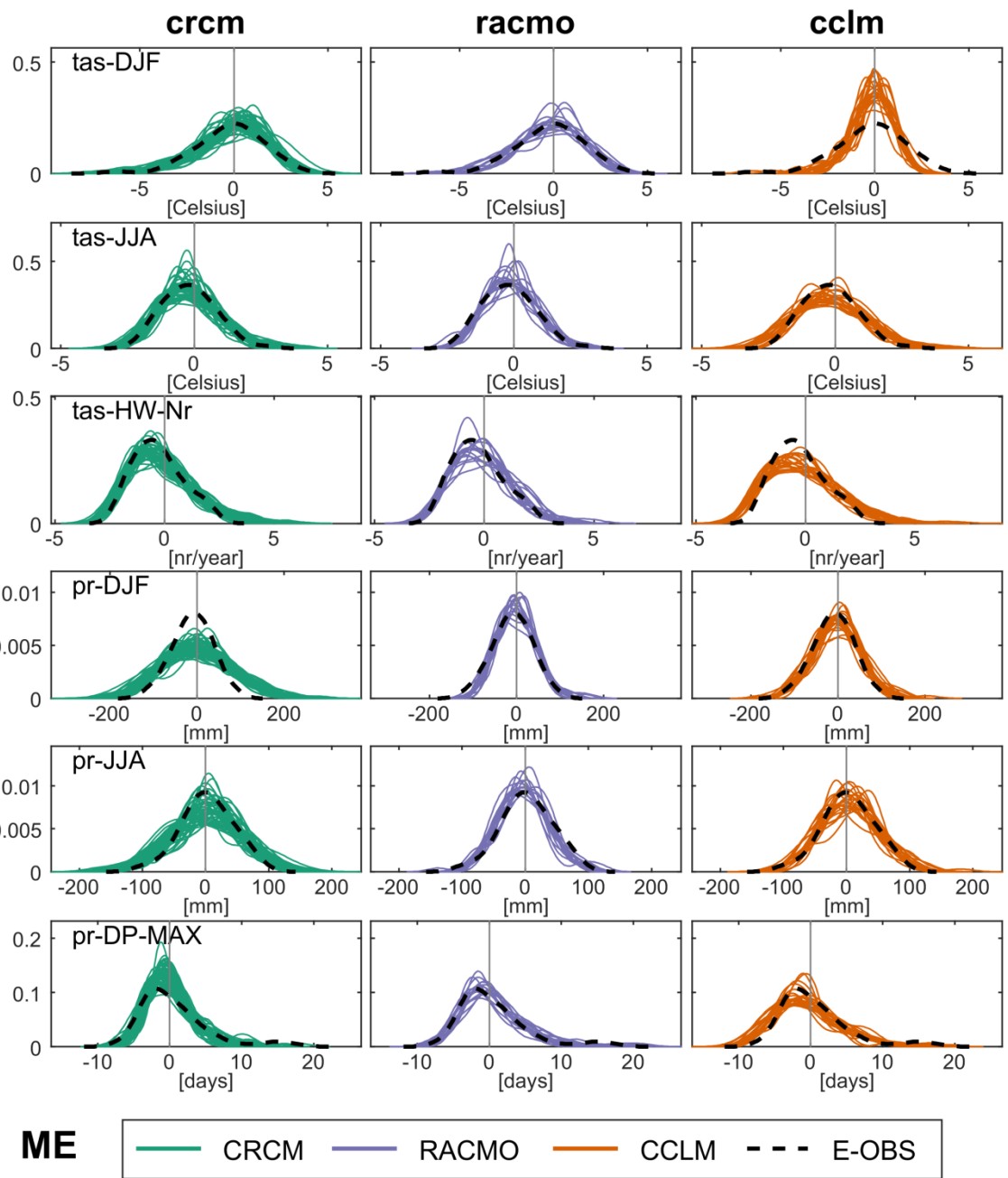

Figure 5: Probability density functions of the annual anomalies during the period 1957-2015 in E-OBS and each ensemble member for all 6 indicators in Mid-Europe (ME)

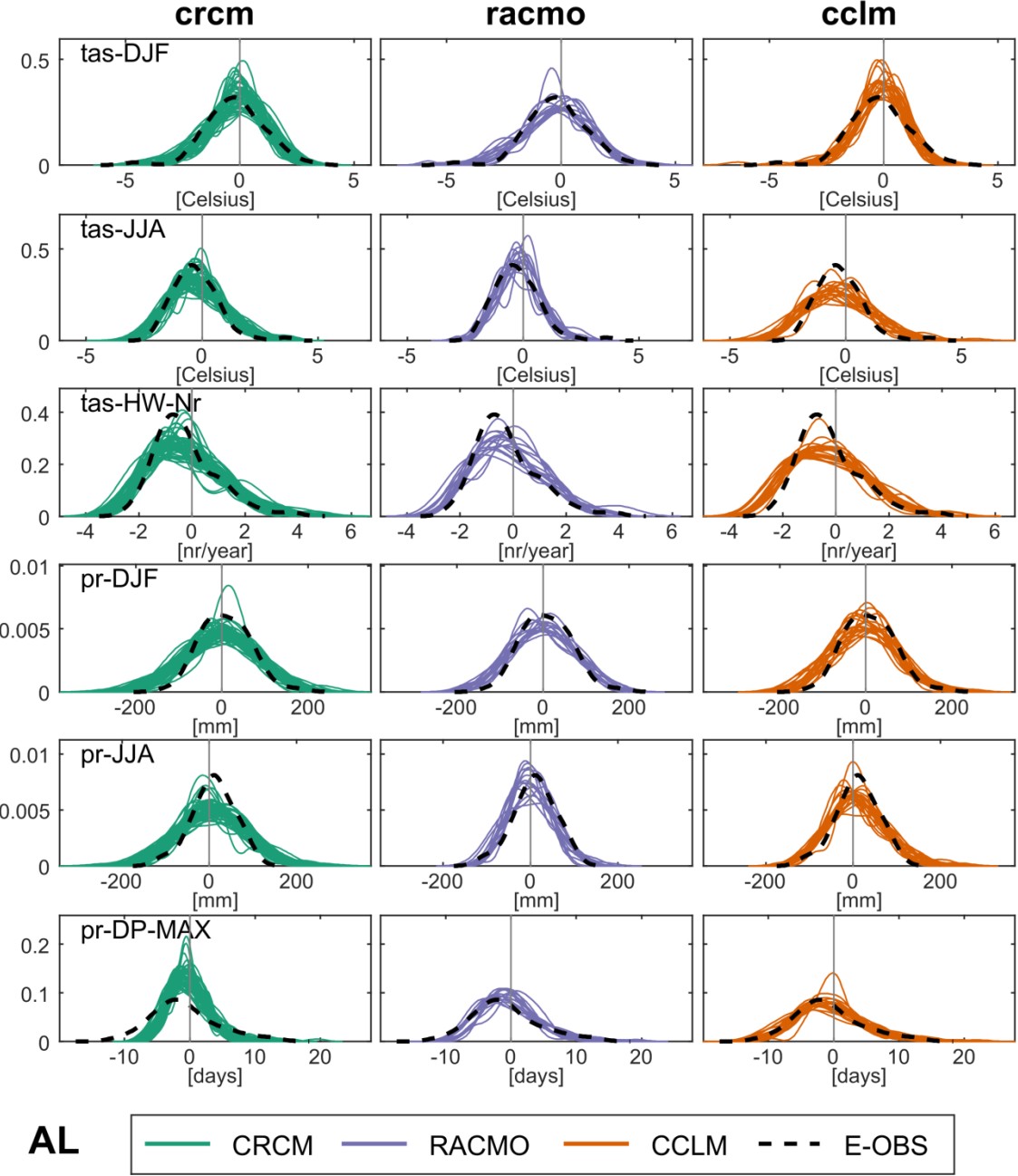

**Figure 6: Probability density functions of the annual anomalies for all 6 indicators in the Alps (AL). For details see Figure 5**

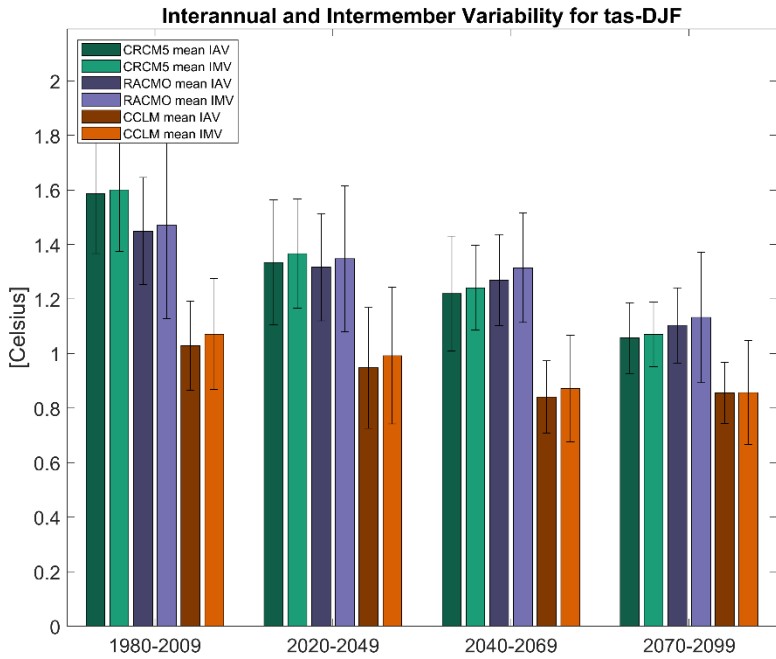

**Figure 7: IAV and IMV of winter temperature in the three ensembles for the reference period (1980-2009) and three future periods. Bars: mean over the variability of each member (IAV) or year (IMV), Error bars: ± standard deviation (members or years); IMV: 16/21/50 members; IAV: 30 years of the respective period**


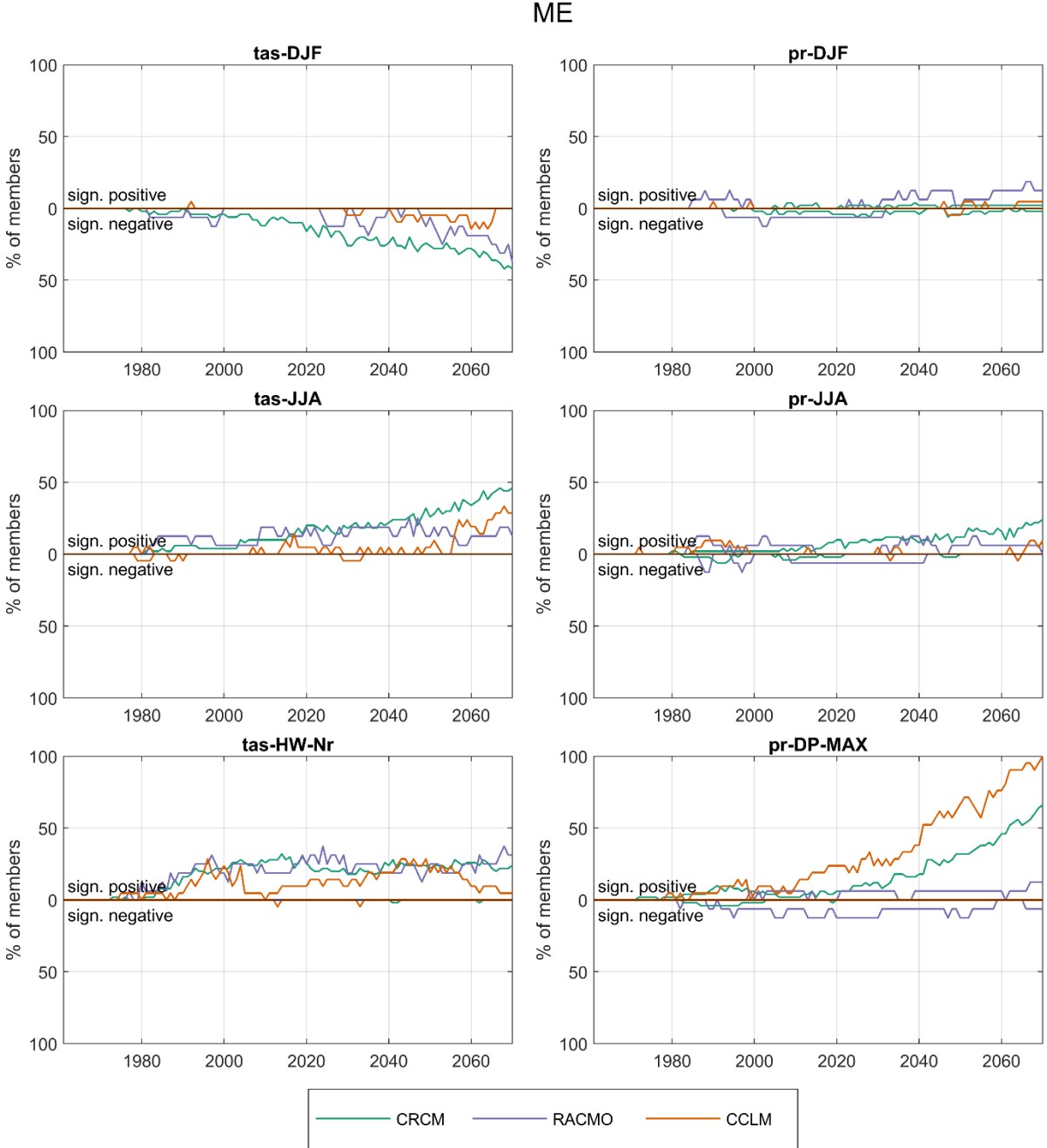

**Figure 8: Percentage of members with significantly different variance (Brown-Forsythe Test with α=0.05) with respect to the reference period 1961-1990 in Mid-Europe. The analysis is based on residuals after removing the EM from each member. The years on the x-axis denote to the starting year of moving 30-year periods.**

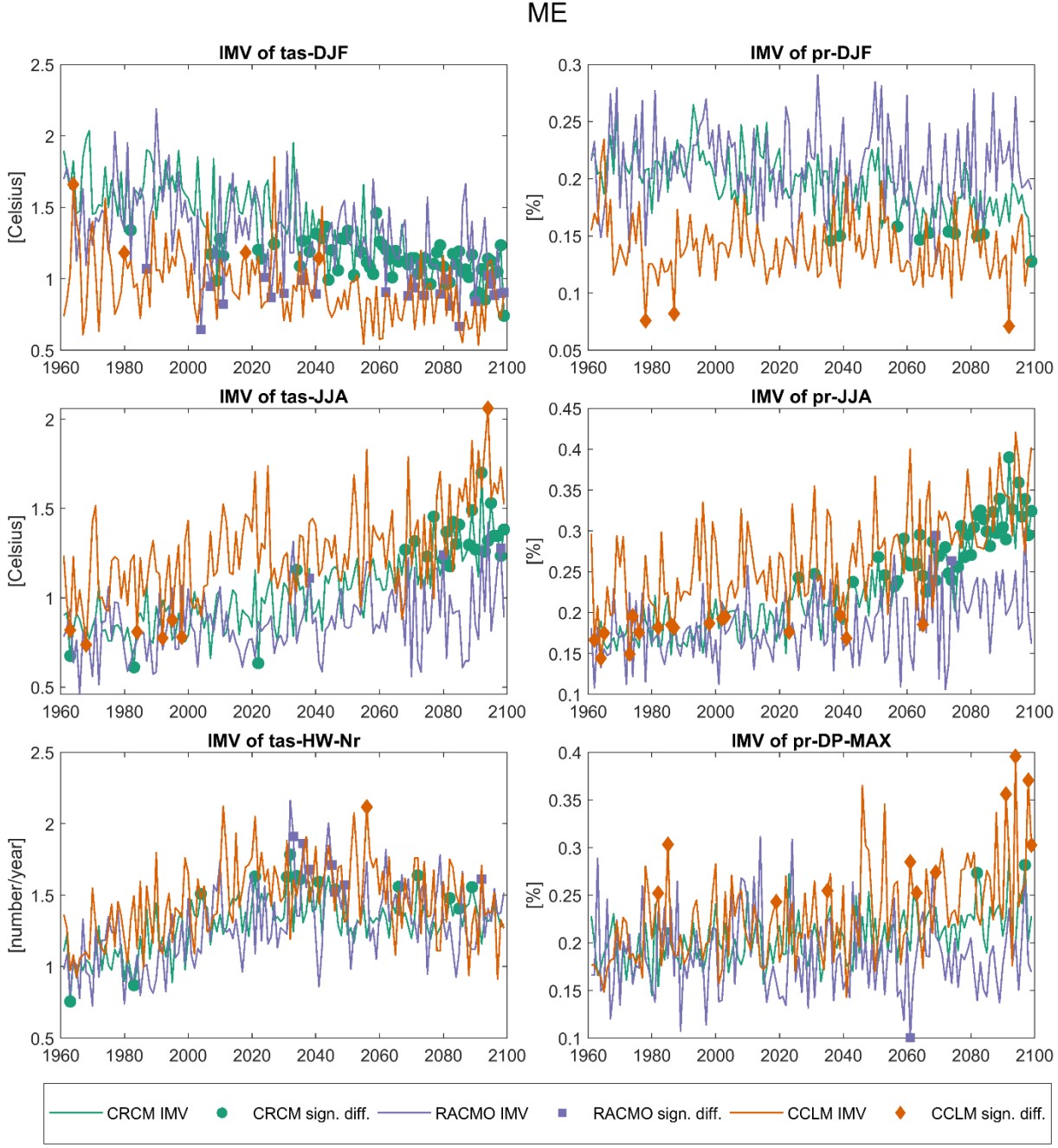

**Figure 9: IMV per year sampled on the dimension of the respective ensemble size (50,21,16) for Mid-Europe. The analysis is based on residuals after removing the EM from each member. The markers highlight years with a significantly different variance than the reference year 1961. Precipitation-based indicators are shown with their relative anomalies from the ensemble mean [percentage].**

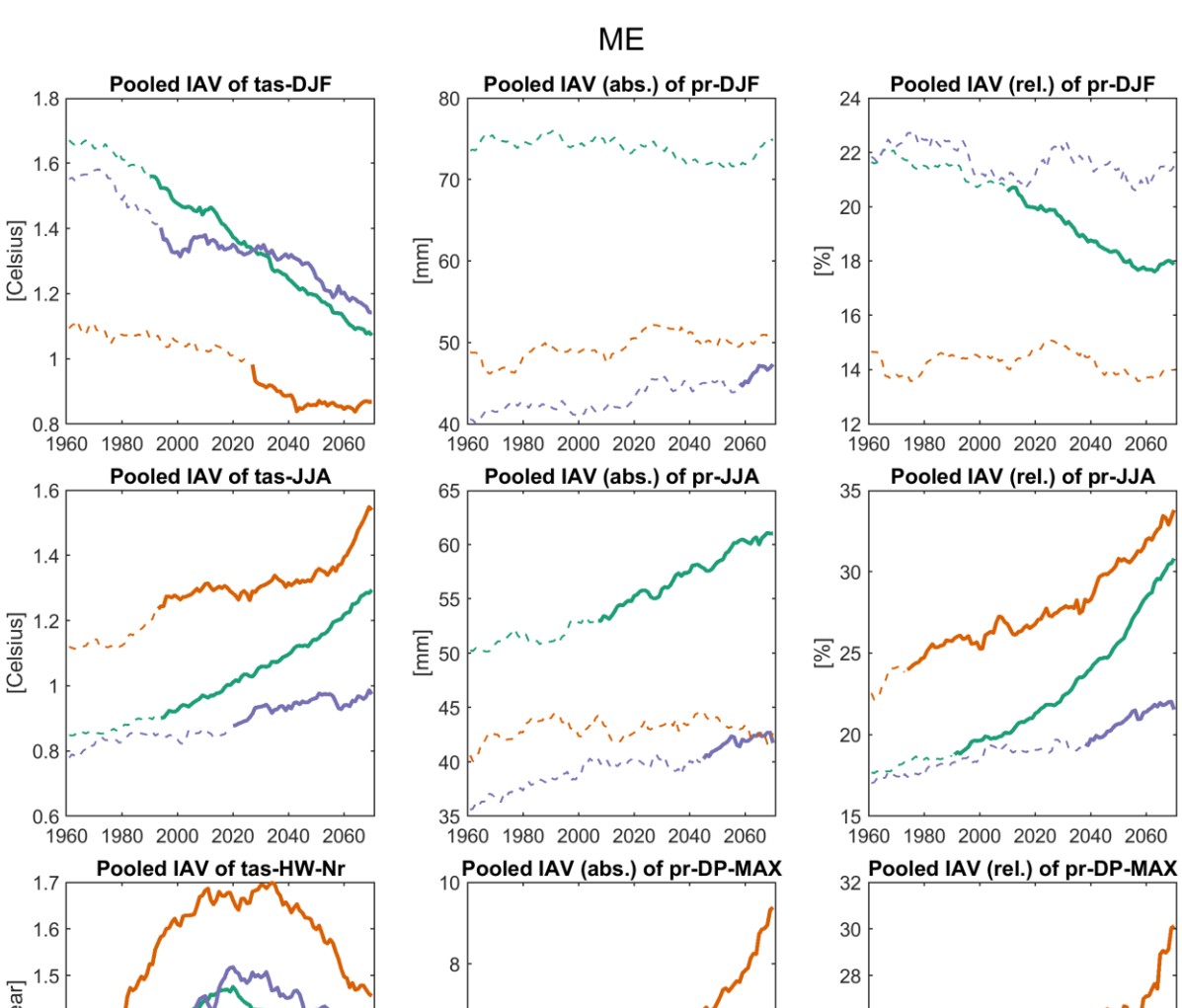

**Figure 10: "Pooled IAV" for Mid-Europe. The analysis is based on residuals, pooled together from all members, after removing the EM from each member. Temperature-based indicators are shown in absolute terms (left column). Precipitation-based indicators are shown both in absolute terms (central column) and relative to the ensemble mean (right column). The change from dashed to solid lines marks the point in time when all following periods show significant changes in variance (BF Test with α=0.0.5).**

**Author contribution**

FvT designed the concept of the study, performed the analysis and created all figures. EA and EF provided the RACMO and CCLM data and helped improving the concept and analysis. FvT led the manuscript writing with input from all authors.

**Referee suggestions**

Flavio Lehner, Jens Christensen, Robert Vautard, Grigory Nikulin

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
