# Peer review of "Comparing internal variabilities in three regional single model initial-condition large ensembles (SMILE) over Europe"

_Earth System Dynamics, 2019_

## Referee Comment (RC1) · Anonymous Referee #1 · 8 Jan 2020

I have difficulties in giving a condense summary of the study by von Trentini and colleagues. As I understand the present ms, the authors intend to investigate the effect of internal variability on projected changes in inter-annual variability of key atmospheric variables over Europe by means of 3 high-resolution SMILEs. This would be scientifically interesting and would provide new and important insights into the uncertainty of simulated future climate change signals. However, the concept of inter-annual variability is treated here as a concept of internal variability (often synonymously), thus causing a lot of confusion of concepts and distraction from a clear line of investigation and argumentation. I don't see an advantage in doing this. Why not investigating projected changes in inter-annual variability the same way as projected changes in any

other variable, using i) the SMILEs to provide a sound estimate of the associated uncertainty due to internal variability (i.e. sensitivity to initial conditions), and ii) the three models to provide an estimate of the associated uncertainty due to the choice of the climate model? The problem becomes apparent already in the introduction (L34ff) when uncertainty in projected climate change signals due to internal variability is confused with "future changes in uncertainty due to internal variability". The mixture of internal and inter-annual variability sometimes leads to unsound comparison, e.g. L34 and 37 (see below), and even to unsound conclusions, e.g. that an increase in inter-annual variability implies an increase in the uncertainty of climate projections due to internal variability (L338).

Moreover, the presented analyses are sometimes questionable. For instance, the significance test for a linear trend in IMV (Fig.9) is based on time series which have been smoothed by a 20-yr running mean. A running mean can heavily reduce the variance of a time series and thus increase the significance of its linear trend artificially. Anyway, the significance of climate change signals is more meaningfully tested against the variance of an unforced control simulation (constant atmospheric greenhouse gas concentrations). Such a control simulation is not provided for any of the RCMs but would be essential for each to substantiate the results.

Further, I miss some important analyses. The results of the investigated RCMs are not compared with their parent driving GCMs (the authors are aware of this, L328). Such a comparison, however, is of high interest and would increase the impact of the study significantly since it allows to assess the error in GCM-based estimates of inter-annual variability. I expect the signal over the British Isles for example to be strongly influenced by the temperature of the ocean, which is prescribed by the GCM in two out of the three RCM ensembles. A RCM-GCM comparison might also provide important information about the influence of the RCM domain size on the projected change signals. The RACMO domain for example is rather small and accordingly I expect a strong influence of the boundary conditions here. Also the imipact of different

ensemble sizes is not discussed but of high interest. The ensemble sizes range from 16 to 50 members. Do the results suggest that 16 ensemble members are enough to study internal and/or inter-annual variability in the atmosphere?

Finally, a great advantage of having 3 ensembles at hand is that we can learn a lot about the driving mechanisms of the simulated future changes and their representations in different models. What are the physical driving mechanisms of the changes that agree in sign and what could be the reasons for the disagreements? The results should be put closer into context e.g. of the studies cited in the introduction (L59-74). Some suggestions of driving mechanisms are already given but should be strengthened by analysing and explaining more details. E.g. L310, arctic amplification and sea ice loss as a driver for decreasing winter temperature variability in Europe is not obvious.

Because of these major concerns, I suggest to reject the ms in its present form. Nevertheless, because of the great potential that I see in the comparison of 3 GCM/RCM SMILEs I like to encourage the authors to revise/extend their study thoroughly and resubmit a new ms.

Other general comments:

It is worth to add to the discussion or conclusions section that the ensemble means of the projected changes can be interpreted as the future changes associated with highest probability (under the considered emission scenario and the individual model constraints) but which specific change would in fact become realized depends on internal variability.

Please also add that by evaluating simulated inter-annual variability with E-OBS you also assume that this single realization (and period) of nature is not an outlier in terms of inter-annual variability under the prevalent climatic conditions.

In many paragraphs, the distinction between historical conditions and projected future

changes is not clear. E.g. L59.

No information about the variations in the initial conditions of both the GCMs and RCMs is provided.

Some specific comments:

14: Suggest: "Simulated inter-annual variability is evaluated against the observational dataset E-OBS and potential future changes under increasing atmospheric greenhouse gas concentrations are compared across the ensembles."

15: Delete sentence "To the knowledge of ..."

34: "Uncertainty of future climate projections can stem from at least three sources ..."

37: In L34, you mention the uncertainty in projected changes due to internal variability. Here you refer to "projected changes in uncertainty" understood as "projected changes in inter-annual variability" which addresses a different aspect of internal variability. These latter projected changes are subject to uncertainty due to internal variability as any other considered variable.

55: Using IMV to quantify IAV should not be motivated by "convenience" but by an advantage. What is the advantage here? Disturbing low-frequency variations are said to be small for seasonal mean and heavy precipitation. What about temperature? Using e.g. a running standard deviation over detrended 30-yr periods would not be sensitive to low-frequency variations. Further, it would be calculated over the same period (30 years) instead of over 16-50 years. IMV is similarly prone to biases due to events in the external forcing.

63: "However" doesn't make sense here.

72: I guess you mean they found significant changes in inter-annual variability only in a small number of CMIP5 models.

118: I assume "surface temperature" refers to 2-m air temperature and "precipitation

sums" to accumulated precipitation. Pleas clarify.

121: Analysis is not limited to summer. A heat wave in winter, though, does not have obvious societal impacts.

140ff: A reference to Fig.5 is missing.

145: "normally" distributed might be more appropriate than "randomly" distributed. The latter more suggests an equal distribution.

159: Why detrended by the ensemble mean rather than by each member individually? The trends are subject to internal variability at lower frequencies and can influence the calculated inter-annual variability.

164: IMV is only insensitive to trends if the trends are the same among the ensemble members. And it is not insensitive to external forcing effects. E.g. if the variability of a specific variable is significantly lower after a volcanic eruption, the IMV would decrease as well. In fact, I would expect the IAV to be generally larger than the IMV (Fig.2). Any idea why IAV < IMV?

218: The E-OBS time series might also be too short to infer a representative pdf, in particular for extremes.

229: Accronyms such as IMV are not used consistently.

234ff: Are these changes significant? For green and blue, the end of the tas-DJF time series shown in Fig.8, for example, seem to be close to or even within the historical ranges shown in Fig.2. This means that the future ranges clearly overlap with the historical ranges. Resting a significance test of a linear trend on smoothed time series, as done in Fig.9, is not valid.

254: Ensemble means are not shown in Fig.5.

258: The correlations shown in Fig.S10/S11 only reflect the signs of the respective changes shown in Fig.5/S2/S4 and do not add any information. In fact, a correlation

analysis between time series subject to trends is heavily influenced by the trends and thus not quite meaningful.

272: Scientific discussions are always critical.

285: Many biases might be inherited from the driving GCMs. A comparison is highly recommended.

290: What is the "coefficient of variation" applied by Giorgi et al.? Why not using it?

294: "Agreement and dissent" evaluates the results as kind of ambivalent. This does not fit with "even better agreement" at the beginning of the next sentence.

304: If I understand the approach correctly, from a future increase in IMV one cannot infer whether this increase is due to an increase in inter-annual variability or due to an increase in the spread of the mean states caused by internal variability. In L55 it is said, that it is valid to use IMV as an approximation for IAV if long-term variations are small compared to IAV. However, long-term variations (including the inter-member spread in the projected change signals) need to be compared with the projected changes in IAV, not only with absolute IAV.

319: Why is it plausible that the statistics of the length of dry periods increase for RCP8.5? In northern Europe, precipitation is projected to increase due to the enhanced moisture transport from low to high latitudes.

329: I highly recommend to include the RCM-GCM comparison in the present study. Whether downscaling with respect to inter-annual variability is important or not can only be demonstrated by such a comparison.

338: I disagree. An increase in inter-annual variability does not imply an increase in the uncertainty of climate projections due to internal variability. Climate change signals are typically based on climatological means. The spread of these is referred to as uncertainty due to internal variability and this metric does not necessarily depend on inter-annual variability.

340: The mean is not shown but required to assess this statement.

---

## Referee Comment (RC2) · Anonymous Referee #2 · 13 Feb 2020

The authors use large ensembles to compare the representation of internal variability in three regional climate models forced by historical and a future scenario forcing. They use observation-based data as a benchmark for the historical period. Large ensemble simulations of single climate models are an essential tool for estimating uncertainty of climate change projections due to internal unforced variability, for detection and attribution studies and so forth. The present study is therefore useful as a validation of such tools and to enhance our understanding of unforced internal climate variability at regional scales. My main concerns with this work however are its presentation, which is confusing at times, the implementation and interpretation of the methodology, and the interpretation of results.

[Figure]

**Major:**

1. The authors recognize that there is confusion in the literature on what is meant by "internal variability" of the climate system (Lines 44-46). I agree. I also agree with the authors' definition of internal variability (Lines 45-51, although it can be shortened). However, in many occasions the authors seem to equate internal unforced variability with inter-annual variability, which add to the confusion (e.g., Lines 10-12, Lines 53-55). I would recommend to clearly define the two from the onset noting that internal variability is unforced whereas inter-annual variability can be externally forced by natural and anthropogenic aerosols, GHGs, solar radiation, land change use and so forth. If inter-annual variability is understood as derived from detrended time series, then explicitly say so from the onset, and clearly indicate how they are detrended.

2. The methodology presented in lines 140–150 is used to asses the interannual variability in the models against that in observations (results in Fig. 5-7). It should be clearly stated that this methodology is not an assessment of model internal (unforced) variability alone, since the time series are affected by the forced signal. Therefore, if there is no agreement between model and observations, we should not conclude that the model representation of internal variability is incorrect, as it may be consequence of the externally forced signal (e.g., the model may have a perfect representation of internal variability, but a too strong response to volcanic eruptions leading to disagreement in the anomaly distributions of Figs. 5-7). On the other hand, I would agree that if the observed and modelled distributions are coincident, this would suggest that both the model response to external forcing and its internal unforced variability are well represented. I don't think this point is clearly made in the methods section and the discussions of sections 4.2 and 5. The way the methodology is presented and the results discussed seem as if the model response to external forcing and that from the observations are in perfect agreement.

3. Based on what is expected from the methodology introduced in lines 140–152, the distributions for the ensemble members in, say, Fig. 7 should largely coincide. They don't. In some cases they are quite different as noted by the authors. It is unclear then how to assess the agreement between model and observations based on these distributions. Are these differences because of a small sample size, or because the ensembles are not large enough? Could they be consequence of poorly sampled (multi)-decadal variability? Can the authors comment on this? I didn't quite follow the rationale of the last sentence of section 4.2, particularly the bit about added "information".

**Minor:**

**In the title:** Consider changing "variabilities" to "variability"

**Line 11:** "... (here: inter-annual variability) ...". Do you mean "on inter-annual timescales"? Inter-annual variability is affected by both externally forced and internally unforced variability. See comments above.

**Lines 53-57:** I don't think this is accurate and should be reworded. The ensemble spread about the mean can be used to measure the internal unforced variability of the model, but may not be representative of inter-annual variations in the presence of, say, a strong volcanic eruption. Therefore, using IMV to assess IAV may be a good approximation in some cases, but may also be in error. This should be clearly stated.

**Line 105:** I believe the work by Fyfe et al., 2017 uses the regional climate model CanESM2-CanRCM4 which is different from CanESM2-CRCM5.

**Line 115:** Although the authors provide a reference, I would find useful a brief comment on the weakness of the E-OBS dataset.

**Line 120-124:** Consider moving the text "These indicators (...) transport of rivers and many more", to the introduction and leave this section only for the methods.

**Lines 129-139:** The discussion on whether the indicators should be computed on the original grid to evaluate the averaged quantities, instead of regridding first and then evaluate the averages over a common grid, seems too long. The authors claim that both approaches give similar results and chose the former, which I believe is the recommended approach (Diaconescu et al., J. Hydrometeorol. 16, 2301–2310). The discussion could be shortened, and this reference cited.

**Line 140–150:** It would be helpful to have explicit references to Figs. 5-7 when discussing the method. Expressions like "we then plot" or "are plotted" are used without showing an actual plot.

**Line 143:** Change "neglected" by "avoided" or the like.

**Line 165:** The authors claim that the spread between members is a well suited metric to examine projected changes of inter-annual variability. I would not agree, unless it refers to detrended inter-annual variability (i.e., after removing the ensemble mean). If so, please clearly specify, although note that the previous sentence (line 163) would then be confusing because both IMV and IAV should be insensitive to external forcing, as no temporal autocorrelation is assumed (line 58).

**Figure 3 and 4:** I'm not sure how to interpret the changes in precipitation. It seems that the plots are intended to show changes in the climatological mean for each ensemble member over two different time periods. Why not showing just that? Precipitation is given here in %. Are these differences of relative values, or relative values of differences? Line 171 defines relative indices by dividing the ensemble anomalies about the mean by the ensemble mean. I can see how this may be useful to assess changes in ensemble spread, but this is not what is

shown in Figs. 3 and 4, right? Please define clearly the quantities in these figures and clearly describe their behaviour.

**Line 202:** Remove "years 1957–2015 in" from the parentheses.

**Line 207:** "(slightly) too low" is odd.

**Line 209:** What does the authors mean by density functions "somewhat inflated"? Please clarify.

**Line 305:** Change to "an statistical model" instead?

---

## Author Comment (AC1) · 30 Apr 2020

**Response to Reviewer Comment #1**

30.04.2020

*First we want to thank you for your long and detailed examination of our manuscript.*
*Some general remarks from our side before we will comment on every paragraph individually:*

*Both reviewers raised concerns on two major aspects:*
*1) confusion about our use of the terms internal variability (IV), inter-annual variability (IAV) and inter-member variability IMV). We carefully define our use of the terms and make the separation clearer in the revised manuscript, and thereby take up your valuable comments on our approach to treat IMV as an approximation for IAV; we still find the concept appealing (and they lead to very similar results as can be seen), but we also understand your concerns about it, especially when it comes to the interpretation of results.*
*2) the lack of a clear line of argumentation. We will stronger focus on the IAV (compared to E-OBS and how it will develop in the future in the 3 SMILEs) throughout the manuscript, and add a chapter on the connections of IAV and IMV at the very end. This controversial topic is thus shifted away from the main line of argumentation.*

*Another important point, especially regarding your comments on further analysis of the driving GCMs and driving mechanisms: The paper is written from an RCM user and impact modeler's perspective, the background of the first author, rather than an atmospheric sciences/model developers perspective. This was obviously not clear enough in the current version and we now emphasize the relevance for the RCM and impact modeler community in the introduction and discussion of the revised manuscript. The revised paper is stronger focused on this perspective, including a more detailed examination of the influence of the model biases in the mean climate state on differences in inter-annual variability in the comparison against E-OBS. This means that we will extend the analysis from the last paragraph of section 5.1 in more depth, as this is very important information for impact modelers, who usually need to bias adjust RCM data.*

I have difficulties in giving a condense summary of the study by von Trentini and colleagues. As I understand the present ms, the authors intend to investigate the effect of internal variability on projected changes in inter-annual variability of key atmospheric variables over Europe by means of 3 high-resolution SMILEs. This would be scientifically interesting and would provide new and important insights into the uncertainty of simulated future climate change signals. However, the concept of inter-annual variability is treated here as a concept of internal variability (often synonymously), thus causing a lot of confusion of concepts and distraction from a clear line of investigation and argumentation. I dont see an advantage in doing this. Why not investigating projected changes in inter-annual variability the same way as projected changes in any other variable, using i) the SMILEs to provide a sound estimate of the associated uncertainty due to internal variability (i.e. sensitivity to initial conditions), and ii) the three models to provide an estimate of the associated uncertainty due to the choice of the climate model? The problem becomes apparent already in the introduction (L34ff) when uncertainty in projected climate change signals due to internal variability is confused with "future changes in uncertainty due to internal variability". The mixture of internal and inter-annual variability sometimes leads to unsound comparison, e.g. L34 and 37 (see below), and even to unsound conclusions, e.g. that an increase in inter-annual variability implies an increase in the uncertainty of climate projections due to internal variability (L338).

*Generally, we will more strictly focus on variability on annual timescales and remove all references to uncertainty in climate projections, which appeared to be confusing. Further clarifications of terms are*

*given in some of the specific comments later.*

*We took up the suggestion to treat IAV just like any other variable and changed the whole methodology for examining future changes in IAV (former Figures 8 and 9). As the new methodology is strongly connected to your comment on lines 234ff ("Are these changes significant? [...]"), it will be presented further down at the respective response. As this is the only part of the manuscript where we actually make use of our assumption that IMV is a good approximation for IAV, all discussion on this point will be shifted to the very end of the manuscript in the revised version. This will make it easier for the reader to follow our line of argumentation. The (obviously) controversial topic can then be discussed in all details without "disturbing" right from the beginning.*

Moreover, the presented analyses are sometimes questionable. For instance, the significance test for a linear trend in IMV (Fig.9) is based on time series which have been smoothed by a 20-yr running mean. A running mean can heavily reduce the variance of a time series and thus increase the significance of its linear trend artificially. Anyway, the significance of climate change signals is more meaningfully tested against the variance of an unforced control simulation (constant atmospheric greenhouse gas concentrations). Such a control simulation is not provided for any of the RCMs but would be essential for each to substantiate the results.

*We agree that the significance test was not a well suited method to test on significant climate changes in IAV. It was therefore discarded. As unforced control simulations are not available for the corresponding RCM simulations, we cannot use them to make statements about the climate change signals compared to pre-industrial times. However we can detect a robust climate change signal in IAV compared to a historical reference climate.*

Further, I miss some important analyses. The results of the investigated RCMs are not compared with their parent driving GCMs (the authors are aware of this, L328). Such a comparison, however, is of high interest and would increase the impact of the study significantly since it allows to assess the error in GCM-based estimates of inter-annual variability. I expect the signal over the British Isles for example to be strongly influenced by the temperature of the ocean, which is prescribed by the GCM in two out of the three RCM ensembles. A RCM-GCM comparison might also provide important information about the influence of the RCM domain size on the projected change signals. The RACMO domain for example is rather small and accordingly I expect a strong influence of the boundary conditions here. Also the imipact of different ensemble sizes is not discussed but of high interest. The ensemble sizes range from 16 to 50 members. Do the results suggest that 16 ensemble members are enough to study internal and/or inter-annual variability in the atmosphere?

*We agree that a comparison of the connections and differences of RCM large ensembles and their driving GCM large ensembles is an important research task. However, such an analysis is out of the scope of this paper. Such an investigation would open up a huge new field of analysis that needs to be performed despite just calculating indices for the GCM data. Since the paper primarily addresses the RCM user/impact modeler community, we kept a GCM-RCM analysis explicitly out of the scope for this paper.*

*Concerning the ensemble size: this is also a very important question that many people ask in the context of SMILEs. We found that the measure we use for the description of variability (standard deviation) is not very sensitive in the range of 16-50 members we have in our data sets. We can add a short section on this. But it is worth mentioning that another paper in this special issue by Sebastian Milinski et al. ("How large does a large ensemble need to be?") is dealing exclusively with the question of ensemble size by means of the 100 member MPI-GE.*

Finally, a great advantage of having 3 ensembles at hand is that we can learn a lot about the driving mechanisms of the simulated future changes and their representations in different models. What are the physical driving mechanisms of the changes that agree in sign and what could be the reasons for

the disagreements? The results should be put closer into context e.g. of the studies cited in the introduction (L59-74). Some suggestions of driving mechanisms are already given but should be strengthened by analysing and explaining more details. E.g. L310, arctic amplification and sea ice loss as a driver for decreasing winter temperature variability in Europe is not obvious.

*We agree that this is an exciting research question. This point is a bit connected to the GCM-RCM discussion, as we also do not see ourselves doing an in depth analysis of the driving mechanisms by means of data analysis. However, there has been a lot of research done towards exploring the mechanisms driving changes in variability. Thus, we extend the discussion of possible driving mechanisms by more explanations and especially with the references to existing literature.*

Because of these major concerns, I suggest to reject the ms in its present form. Nevertheless, because of the great potential that I see in the comparison of 3 GCM/RCM SMILEs I like to encourage the authors to revise/extend their study thoroughly and resubmit a new ms.

*With a stronger emphasis on the key points we want to analyze in the paper, clearer distinction between inter-annual variability and uncertainty due to internal variability, and the changes in methodology already (and later on) mentioned we hope to convince you with a more concise and comprehensible manuscript, following a clear line of argumentation.*

Other general comments:

It is worth to add to the discussion or conclusions section that the ensemble means of the projected changes can be interpreted as the future changes associated with highest probability (under the considered emission scenario and the individual model constraints) but which specific change would in fact become realized depends on internal variability.

*We will pick up this comment and add a sentence or two to clarify this important point.*

Please also add that by evaluating simulated inter-annual variability with E-OBS you also assume that this single realization (and period) of nature is not an outlier in terms of inter-annual variability under the prevalent climatic conditions.

*We will discuss this. Note that we evaluate whether IAV in E-OBS falls within the range of IAV in any of the ensemble members. Of course the observed variability could still result from a realization of climate that is an outlier but in contrast to most previous studies based on individual simulations we consider our analysis as more robust.*

In many paragraphs, the distinction between historical conditions and projected future changes is not clear. E.g. L59.

*We will go through the whole manuscript to always make clear what the subject of discussion in the respective part is.*

*No information about the variations in the initial conditions of both the GCMs and RCMs is provided.*

*We did not include this in the original manuscript as it is already documented in the cited literature. According to the reviewer's suggestion we now summarize the initialization techniques in the revised manuscript.*

Some specific comments:

14: Suggest: "Simulated inter-annual variability is evaluated against the observational dataset E-OBS and potential future changes under increasing atmospheric greenhouse gas concentrations are compared across the ensembles."
*we will change this sentence*

15: Delete sentence "To the knowledge of ..."
*Changed accordingly*

34: "Uncertainty of future climate projections can stem from at least three sources ..."
*Changed accordingly*

37: In L34, you mention the uncertainty in projected changes due to internal variability. Here you refer to "projected changes in uncertainty" understood as "projected changes in inter-annual variability" which addresses a different aspect of internal variability. These latter projected changes are subject to uncertainty due to internal variability as any other considered variable.
*As mentioned in the general comments at the beginning, we will clarify our use of terminology and how they are connected more properly; especially the differences between internal variability, stemming from initial conditions and the inter-annual variability; this includes the usage of "uncertainty" as in this case*

55: Using IMV to quantify IAV should not be motivated by "convenience" but by an advantage. What is the advantage here? Disturbing low-frequency variations are said to be small for seasonal mean and heavy precipitation. What about temperature? Using e.g. a running standard deviation over detrended 30-yr periods would not be sensitive to low-frequency variations. Further, it would be calculated over the same period (30 years) instead of over 16-50 years. IMV is similarly prone to biases due to events in the external forcing.
*The advantage of using IMV as an estimate for unforced IAV is particularly relevant in the presence of non-linear forcing. For instance as pointed out by reviewer #2 the response to a volcanic eruption cannot be easily separated from unforced IAV. Also the anthropogenic forcing since 1950 has not been linear in time. IMV is an elegant way to get around this challenge. Since we use standard deviation as a metric it is also not sensitive to the use of 16, 30 or 50 years. The approach of using IMV as an approximation of IAV has been used in several previous studies, e.g. in Leduc et al. (2019, p. 681), where the authors state that "In the case of a climate system under transient forcing, the use of Eq. (1) to assess temporal variability using the inter-member spread involves weaker assumptions than calculating the residual temporal variability from detrended time series.", based on a study by Nikiéma, Laprise et al. (2018) [DOI 10.1007/s00382-017-3918-0].*

63: "However" doesnlt make sense here.
*removed it*

72: I guess you mean they found significant changes in inter-annual variability only in a small number of CMIP5 models.
*Exactly. We will change the sentence to make it clear.*

118: I assume "surface temperature" refers to 2-m air temperature and "precipitation sums" to accumulated precipitation. Pleas clarify.
*In two ensembles the variable is specified as "near-surface air temperature" and in one ensemble as "2m air temperature" – we will change the terminology to be more accurate. Although the term "precipitation sum" seems to be as unambiguous as "accumulated precipitation", we will change that.*

121: Analysis is not limited to summer. A heat wave in winter, though, does not have obvious societal impacts.

*The heatwaves are here defined as consecutive days above the 95th percentile calculated across all days of the year and not as time varying percentiles as in some other definitions. We can expect that all these days occur during the summer months. Experiencing this during winter seems very unlikely, even under RCP8.5. And even if so, such high temperatures (assume a summer heatwave as seen in the current climate) would probably have even worse impacts when occurring in winter, especially for ecosystems.*

140ff: A reference to Fig.5 is missing.
*added*

145: "normally" distributed might be more appropriate than "randomly" distributed. The latter more suggests an equal distribution.
*true, changed accordingly*

159: Why detrended by the ensemble mean rather than by each member individually? The trends are subject to internal variability at lower frequencies and can influence the calculated inter-annual variability.
*We aim at estimating the total unforced variability at interannual time scales including the contribution from low frequency variability. When removing a linear trend from each individual realization we remove some of this variability e.g. if the first or last year is extremely warm or cold. The multi member trend is a much better estimate of the forced response than each member's trend.*

164: IMV is only insensitive to trends if the trends are the same among the ensemble members. And it is not insensitive to external forcing effects. E.g. if the variability of a specific variable is significantly lower after a volcanic eruption, the IMV would decrease as well. In fact, I would expect the IAV to be generally larger than the IMV (Fig.2). Any idea why IAV < IMV?
*The first idea we had was that it is because the IAV data is detrended, while the data for IMV is the original values. However, after looking at the methodology again, we found that both IAV and IMV are based on the ensemble-mean-detrended time series for this comparison. Right now, we cannot explain why there is a systematic bias towards higher values of IMV, but we hope to find some explanation for the revised manuscript. This will be part of the extended discussion on the similarities and differences between IAV and IMV.*

218: The E-OBS time series might also be too short to infer a representative pdf, in particular for extremes.
*We will generally emphasize the limitations of an observed reference in this context, as it is always just one realization*

229: Accronyms such as IMV are not used consistently.
*Changed accordingly*

234ff: Are these changes significant? For green and blue, the end of the tas-DJF time series shown in Fig.8, for example, seem to be close to or even within the historical ranges shown in Fig.2. This means that the future ranges clearly overlap with the historical ranges. Resting a significance test of a linear trend on smoothed time series, as done in Fig.9, is not valid.
*Thank you very much for this useful comment. We took it up and totally changed the analysis of future changes in IAV (former figures 8 and 9). The new methodology, based on Brown-Forsysth tests (which is less susceptible to non-normality than a regular F-Test) for equal variances is introduced now:*
*Overlapping 30-year periods that are shifted by one year each are the basis (1961-1990, 1962-1991, …, 2070-2099) to detect changes in the variance of annual data (detrended by the ensemble mean) for each indicator. A Brown-Forsythe Test on equal variances (alpha=0.05) is performed for each of the periods with respect to the first period that serves as a reference (1961-1990). This is done for*

*each member individually. We thus get information on changes in the variance for each period and each member. Together with the sign of change in variance in each of these cases, we can extract the number of members per period that show a statistically significant change (positive or negative) in variance (Figure 1). Results show that only a minority of members shows significant changes of variance. For the example of tas-DJF (ME), all members show a decrease in IAV, calculated as standard deviation of detrended 30-year periods (Figure 2). However this relatively clear change in the metric "standard deviation" does not imply significance in all members, as you stated correctly. This is what we can now show with the improved methodology, and this also exceeds much of the analysis in existing literature. Here, often no significance test of detected changes in inter-annual variability is performed. Many studies just inform about the robustness of change (e.g. by stippling in maps), measured by the accordance of (usually) 67% of the models of multi-model ensembles. This does however not allow information about the significance compared to a reference climate.*

[Figure]

*Figure 1: Percentage of members showing a significant change of variance in the respective period against the reference period 1961-1990 for the region ME. The x-axis depicts the starting year of each 30-year period. Positive changes are shown upwards, while negative changes are shown downwards.*

[Figure]

*Figure 2: Temporal evolution of IAV (measured as standard deviation across 30-year periods, detrended with the ensemble mean) for the 50 members of CRCM fir tas-DJF in ME*

Additional to this analysis of changes in IAV itself, we still think the concept of IMV as an approximation for IAV is useful, as it can incorporate both the information on the general direction of change as well as the significance of these changes. We therefore perform a similar methodology as for IAV on the IMV data: in contrast to IAV, IMV gives one value per year that we can easily plot for each SMILE. We then apply the Brown-Forsysth Test on the variances of all members per year to detect significant changes in the variance. This is done for all years with respect to the variance of the first year (1961). The results are shown in Figure 3 for ME. In only a very few cases, the IMV change is significant (solid line type). Why the IMV shows increasing tendencies for tas-JJA, while there are a rather negative tendencies in the IAV (Figure 1), needs to be clarified for the revised version. It might well be that this inconsistency exists because the IMV is based on the original data, not on detrended time series as used for the calculation of IAV in Figure 1. Unfortunately we were not able to perform the analysis with detrended data for the IMV data of Figure 3 until the deadline for this response. We will handle this in the new chapter, where all issues concerning the connections between IAV and IMV will be discussed.

ME

[Figure]

*Figure 3: Temporal evolution of IMV per year. For each year the significance of change is evaluated with a Brown-Forsyth Test on equal variances against the first year (1961). The line style changes from dashed to solid when all following years show a significantly different variance, indicating a consistent change in variability.*

254: Ensemble means are not shown in Fig.5.
*correct, we will add lines to show the ensemble means*

258: The correlations shown in Fig.S10/S11 only reflect the signs of the respective changes shown in Fig.5/S2/S4 and do not add any information. In fact, a correlation analysis between time series subject to trends is heavily influenced by the trends and thus not quite meaningful.
*These figures and their discussion will be removed*

272: Scientific discussions are always critical.
*of course*

285: Many biases might be inherited from the driving GCMs. A comparison is highly recommended.
*As mentioned above, while a comparison is highly interesting, it is outside the scope of the current manuscript.*

290: What is the "coefficient of variation" applied by Giorgi et al.? Why not using it?

*The coefficient of variation is defined as the ratio of std/mean, which is the same as using the standard deviation of relative differences from the mean, as we did; so the result is the same, although slightly different calculations were performed. We mention the work by Giorgi et al. to add a reference to similar thoughts around the interpretation of variability for precipitation based indicators. This will be clearer in the revised manuscript.*

294: "Agreement and dissent" evaluates the results as kind of ambivalent. This does not fit with "even better agreement" at the beginning of the next sentence.

*we will improve the wording in this section by discarding the "even"*

304: If I understand the approach correctly, from a future increase in IMV one cannot infer whether this increase is due to an increase in inter-annual variability or due to an increase in the spread of the mean states caused by internal variability. In L55 it is said, that it is valid to use IMV as an approximation for IAV if long-term variations are small compared to IAV. However, long-term variations (including the inter-member spread in the projected change signals) need to be compared with the projected changes in IAV, not only with absolute IAV.

*Note that ideally IAV in transient simulations could be calculated over a very long time period of e.g. more than 100 years and thereby it would also include a contribution from low frequency variability. Note that we do not explicitly use a high-pass filter here to calculate IAV. However, we here calculate IAV over a detrended 30-yr period. Thereby our calculation of IAV does not account for low frequency internal variability, which is why we stated that it is valid to use our IMV calculation as an approximation of IAV.*

*Likewise, the change in future IMV may arise from changes in low-frequency variability (which would lead to a larger spread in 30-yr means) or high-frequency variability e.g., IAV within 30 year periods. Thus we compare in the revised version changes in IMV and IAV calculated over 30-yr periods.*

319: Why is it plausible that the statistics of the length of dry periods increase for RCP8.5? In northern Europe, precipitation is projected to increase due to the enhanced moisture transport from low to high latitudes.

*Increases in the length of dry periods do not necessarily contradict increasing precipitation in general; CRCM5 also shows an increase for the ensemble mean of pr-DP-MAX for Scandinavia from 18 to 20 days, although the IMV is quite large (6-8 days), why this change is probably not significant. However, EURO-CORDEX data from Jacobs et al. (2014) do not show significant increases for the length of dry spells in Northern Europe. We will change this sentence accordingly, also differing between heatwaves and dry spells.*

329: I highly recommend to include the RCM-GCM comparison in the present study. Whether downscaling with respect to inter-annual variability is important or not can only be demonstrated by such a comparison.

*see above*

338: I disagree. An increase in inter-annual variability does not imply an increase in the uncertainty of climate projections due to internal variability. Climate change signals are typically based on climatological means. The spread of these is referred to as uncertainty due to internal variability and this metric does not necessarily depend on inter-annual variability.

*As stated above, we will better disentangle the two terms and adapt the conclusions drawn. We agree that an increase in IAV does not necessarily imply an increase in uncertainty of climate projections due to internal variability.*

340: The mean is not shown but required to assess this statement.
*added*

---

## Author Comment (AC2) · 30 Apr 2020

30.04.2020

*First we want to thank you for your long and detailed examination of our manuscript.*
*Some general remarks from our side before we will comment on every paragraph individually:*

*Both reviewers raised concerns on two major aspects:*
*1) confusion about our use of the terms internal variability (IV), inter-annual variability (IAV) and inter-member variability IMV). We carefully define our use of the terms and make the separation clearer in the revised manuscript, and thereby take up your valuable comments on our approach to treat IMV as an approximation for IAV; we still find the concept appealing (and they lead to very similar results as can be seen), but we also understand your concerns about it, especially when it comes to the interpretation of results.*
*2) the lack of a clear line of argumentation. We will stronger focus on the IAV (compared to E-OBS and how it will develop in the future in the 3 SMILEs) throughout the manuscript, and add a chapter on the connections of IAV and IMV at the very end. This controversial topic is thus shifted away from the main line of argumentation.*

The authors use large ensembles to compare the representation of internal variability in three regional climate models forced by historical and a future scenario forcing. They use observation-based data as a benchmark for the historical period. Large ensemble simulations of single climate models are an essential tool for estimating uncertainty of climate change projections due to internal unforced variability, for detection and attribution studies and so forth. The present study is therefore useful as a validation of such tools and to enhance our understanding of unforced internal climate variability at regional scales. My main concerns with this work however are its presentation, which is confusing at times, the implementation and interpretation of the methodology, and the interpretation of results.

Major:

1. The authors recognize that there is confusion in the literature on what is meant by "internal variability" of the climate system (Lines 44-46). I agree. I also agree with the authors' definition of internal variability (Lines 45-51, although it can be shortened). However, in many occasions the authors seem to equate internal unforced variability with inter-annual variability, which add to the confusion (e.g., Lines 10-12, Lines 53-55). I would recommend to clearly define the two from the onset noting that internal variability is unforced whereas inter-annual variability can be externally forced by natural and anthropogenic aerosols, GHGs, solar radiation, land change use and so forth. If inter-annual variability is understood as derived from detrended time series, then explicitly say so from the onset, and clearly indicate how they are detrended.
*We agree that IMV of annual data might differ from IAV in the presence of forced changes due to aerosols, GHGs, solar and volcanic forcing. In the revised manuscript we will state more clearly that IMV of annual data is used as (a very good)* estimate *of IAV. Indeed, we determine IAV in observations and simulations as the standard deviation of detrended time series. We will clarify the terms and their usage in our study throughout the whole manuscript. We took the approach of using IMV of annual data as an approximation of IAV from Leduc et al. (2019, p. 681), where the authors state that "In the case of a climate system under transient forcing, the use of Eq. (1) to assess temporal variability using the inter-member spread involves weaker assumptions than calculating the residual*

*temporal variability from detrended time series.", based on a study by Nikiéma, Laprise et al. (2018) [DOI 10.1007/s00382-017-3918-0]. In the current version of the manuscript we show that IMV is indeed a good estimate for IAV in the reference period. In the revised manuscript we will also include a comparison of future changes in IMV of annual data and IAV calculated from 30-year periods.*

2. The methodology presented in lines 140–150 is used to asses the interannual variability in the models against that in observations (results in Fig. 5-7). It should be clearly stated that this methodology is not an assessment of model internal (unforced) variability alone, since the time series are affected by the forced signal. Therefore, if there is no agreement between model and observations, we should not conclude that the model representation of internal variability is incorrect, as it may be consequence of the externally forced signal (e.g., the model may have a perfect representation of internal variability, but a too strong response to volcanic eruptions leading to disagreement in the anomaly distributions of Figs. 5-7). On the other hand, I would agree that if the observed and modelled distributions are coincident, this would suggest that both the model response to external forcing and its internal unforced variability are well represented. I don't think this point is clearly made in the methods section and the discussions of sections 4.2 and 5. The way the methodology is presented and the results discussed seem as if the model response to external forcing and that from the observations are in perfect agreement.
*We will take up these valuable comments in the methods and discussion section. As written above, we plan to include a section on the biases of the models. These comments can be an integral part of this extended section.*

3. Based on what is expected from the methodology introduced in lines 140–152, the distributions for the ensemble members in, say, Fig. 7 should largely coincide. They don't. In some cases they are quite different as noted by the authors. It is unclear then how to assess the agreement between model and observations based on these distributions. Are these differences because of a small sample size, or because the ensembles are not large enough? Could they be consequence of poorly sampled (multi-)decadal variability? Can the authors comment on this? I didn't quite follow the rationale of the last sentence of section 4.2, particularly the bit about added "information".

The fact that different members disagree on the distribution of the annual anomalies (internal variability) is exactly the reason why large ensembles are required to accurately estimate inter-annual variability and changes therein. We expect the differences indeed to mainly originate from the sample size (30 years), but *low frequency variations that differ for the 30 years of the reference period between members, with which the members are normalized, may play a role as well. The last sentence of section 4.2 tries to comment on the occurrence of these outlier members. They have a distinctly different distribution than all other members, just because of their initial conditions (this is meant by "how large the influence of internal variability can be…"). The statement on "added information" points to the question of the size of SMILEs needed to get a satisfying quantification of internal variability (thus, how much results differ from member to member). These outlier members can "add information" on the magnitude of internal variability even after 49 relatively similar members. Let's pretend you take a look at these distributions one member after another and the outlier is the last one. You would guess that the distributions of the 49 members you had seen so far give you a good approximation of how further members' distributions would look like. However, the last one looks quite different, thus "adding information". We will make this clearer in the revised manuscript.*

Minor:

In the title: Consider changing "variabilities" to "variability"
*changed accordingly*

Line 11: "... (here: inter-annual variability) ...". Do you mean "on inter-annual timescales"? Inter-annual variability is affected by both externally forced and internally unforced variability. See comments above.
*We will rephrase this sentence of the abstract according to a generally better definition of terms.*

Lines 53-57: I don't think this is accurate and should be reworded. The ensemble spread about the mean can be used to measure the internal unforced variability of the model, but may not be representative of inter-annual variations in the presence of, say, a strong volcanic eruption. Therefore, using IMV to assess IAV may be a good approximation in some cases, but may also be in error. This should be clearly stated.
*We will more clearly state that IMV on annual data is an estimate of IAV in the revised manuscript.*

Line 105: I believe the work by Fyfe et al., 2017 uses the regional climate model CanESM2-CanRCM4 which is different from CanESM2-CRCM5.
*Fyfe et al was used as a source for information on the driving CanESM2 data, describing the initialization and creation of members. More recent papers seem to be using Kirchmeier-Young et al (2017) [10.1175/JCLI-D-16-0412.1] for a description of the CanESM2 LE, so we will exchange the source.*

Line 115: Although the authors provide a reference, I would find useful a brief comment on the weakness of the E-OBS dataset.
*We will include a short comment.*

Line 120-124: Consider moving the text "These indicators (...) transport of rivers and many more", to the introduction and leave this section only for the methods.
*Our general plan for the revised version is to adjust the storyline of the paper more to an RCM users' and impact modelers' perspective. We will therefore extent the part on impact-relevant information and, move them to the introduction and pick them up in the conclusions part.*

Lines 129-139: The discussion on whether the indicators should be computed on the original grid to evaluate the averaged quantities, instead of regridding first and then evaluate the averages over a common grid, seems too long. The authors claim that both approaches give similar results and chose the former, which I believe is the recommended approach (Diaconescu et al., J. Hydrometeorol. 16, 2301–2310). The discussion could be shortened, and this reference cited.
*Thank you for the hint to this publication.*

Line 140–150: It would be helpful to have explicit references to Figs. 5-7 when discussing the method. Expressions like "we then plot" or "are plotted" are used without showing an actual plot.
*we will add references*

Line 143: Change "neglected" by "avoided" or the like.
*we will*

Line 165: The authors claim that the spread between members is a well suited metric to examine projected changes of inter-annual variability. I would not agree, unless it refers to detrended inter-annual variability (i.e., after removing the ensemble mean). If so, please clearly specify, although note that the previous sentence (line163) would then be confusing because both IMV and IAV should be insensitive to external forcing, as no temporal autocorrelation is assumed (line 58).
*The time series are indeed detrended with the ensemble mean as stated right before in line 159. In*

*the revised manuscript we will more clearly define IMV and IAV and their characteristics, as discussed above.*

Figure 3 and 4: I'm not sure how to interpret the changes in precipitation. It seems that the plots are intended to show changes in the climatological mean for each ensemble member over two different time periods. Why not showing just that? Precipitation is given here in %. Are these differences of relative values, or rel- ative values of differences? Line 171 defines relative indices by dividing the en- semble anomalies about the mean by the ensemble mean. I can see how this may be useful to assesschanges in ensemble spread, but this is not what is shown in Figs. 3 and 4, right? Please define clearly the quantities in these figures and clearly describe their behaviour.

*You are right, they just show the differences between the mean states of two periods. The change is given in % for each member, thus an increase from 100mm to 120mm will result in a value of 20% for that member. The definition from line 171 is referring to the change in internal variability in the future (Fig. 8). We will add more information to the figure description, and in the methods part, where it is indeed missing at the moment.*

Line 202: Remove "years 1957–2015 in" from the parentheses.
*probably redundant, yes*

Line 207: "(slightly) too low" is odd.
*we will change the wording here*

Line 209: What does the authors mean by density functions "somewhat inflated"? Please clarify.
*The comparison between observations and simulations is based on the original data (opposed to detrended data). The total variability shown here thus includes the inter-annual variability around the mean (IAV is defined as the variability of detrended time series) and the deviation due to a trend in the mean climate state, i.e. the variability could be somewhat larger than IAV calculated from detrended time series.*

Line 305: Change to "an statistical model" instead?
*that is shorter indeed*

---

## Author Response (AR2)

**REPLY TO REVIEWER #3 for the revised manuscript**

*We want to thank the reviewer for her/his valuable comments and suggestions. Suggestions that were directly adapted are marked in green, while additional replies from the authors are marked in blue.*

This paper compares projections of seasonal temperature, precipitation, heatwaves and droughts over Europe on three SMILEs based on regional climate models. The paper has some really interesting results, such as the general agreement of the sign of the change, although not the magnitude across the three SMILEs used. The paper provides a significant contribution to the literature as possibly the first intercomparion of regional climate model SMILEs. The scientific results are generally sound and well founded, however the paper itself often lacks clarity and precision, making the job of the reader difficult at times. I recommend revisions to the text in the manuscript before publication.

This is my first review of the manuscript. I note that the following issue pointed out by the original reviewers is not completely resolved.

1. There are still some instances where the term internal variability is confusing and it is unclear which metric is being talked about and whether you refer to inter-annual variability or general variability. We went through the text and clarified all occurrences of "internal variability" and "variability"

Examples are:
lines 224/234

Comments on the manuscript:

Section 3.2 – can you do a statistical test on the means and spread to be able to say for certain whether the different models are truly different? We added some information to section 3.2 concerning the statistical tests for mean (T-test) and variance (Brown-Forsythe Test)

91-110 – should this explanation be in the introduction or methods? This is a valid comment as it would also fit into the methods section. We chose to leave it in the introduction, since it sets the frame for all coming sections and there is no proper place for these paragraphs in the methods section

164/174 – is this really all models? Or all ensemble members? This is an important distinction The figures clearly show that it is both why do we expect this to be normally distributed? Because the "perfect" model would totally capture the IAV as IMV, and IAV of precipitation should be normally distributed in Europe

– you say 'looks like' can you test this? It would be possible to test the observational distribution against each member individually, using a KS test for example. If all individual tests would show no similarity, one could state that it is rather unlikely that the observations could be part of the ensemble of distributions from the respective model. It would, however, still only allow us to make a probabilistic statement – we cannot say for sure if the model is able to represent observational IAV satisfactorily (see discussion on outlier members). The general problem that we only have one realization of historical climate (variability) remains, why we chose to leave this a qualitative analysis.

In general I am confused about how you calculate some of the variability metrics, in some places you refer to standard deviations and other places variance. Can you clarify this?
Lines 91, 234,235,239,246 are places where I was unsure what metric you were using IAV is calculated as standard deviations, while the BF Test checks for equal variances. Since std is the square root of var, and they are therefore strongly tied, we use both metrics. We went through the text and made changes where necessary.

– can you be more specific about the 'period around this year' statement? We added some information

Conclusions: These are mostly context and future work. Can you make some stronger conclusions from this manuscript. What did you find? Why is it important? We do not agree. We clearly state what we found (e.g. new lines 458-463) and say why this is important (454-457, 463-464, last two paragraphs)

The following set of textual suggestions are meant as just suggestions. These are some examples of where the choice of wording is confusing or the grammer is incorrect. This is not an extensive list and I strongly suggest the authors revisit the text for clarity and to make sure all sentences are gramatically correct and make sense.

Textual suggestions (with line number first):

Abstract
Define IAV in its first instance
get rid of 'often'
might be better to say something like: 'represent a combination of external forcing and IAV'
powerful 'tools'
'many' instead of 'multiple'
'summer and winter'
what do you mean by the two extreme indicators? How are they biased? The two extreme indicators are the number of heatwaves and the maximum length of dry periods – in contrast to seasonal tas and pr we think it is clear that these can be considered extreme indicators. The biases do not really show a systematic behavior, why we did not specify this in the abstract. A new wording tries to make it clearer what we found without making this information too long for the abstract, as it is not one of the main findings of the paper.
what do you mean 'mostly follow'? we changed the wording to 'also show'
'some of the individual realisations show' the current wording better emphasizes the differences between realizations and is therefore not changed
'the' significance
dont need both 'further' and 'also'

Introduction
remove the words 'Next to' replace with 'In addition to' and replace 'another' with 'an'
'on' different timescales
replace 'like' with 'such as'
'utmost'
replace 'especially' with 'particularly'
do you need tas/pr in the intro, I would move this information to the methods We want to keep it here where all indicators are listed in the main text the first time

Data
'between' the three ensembles
'with' different
'each initialisation using atmospheric perturbations'
replace inter-compared with 'compared'
'because of its availability for Europe and because it has similar spatial resolution'

Section 3.1
You first words are 'The indicators' - the reader does not remember what this refers to We added the term indicators in the (new) lines 39/40 and refer to Table 1 at the beginning of section 3.1

Section 3.2
replace 'show' with precipitation increase in all models – models don't show a result they do something
similar standard deviation and range of what? Of changes in pr and tas; we added this to the text

Section 3.3
sentence is unclear we changed it to clarify it
should read 'is concentrated inside'
– stable conditions is unclear changed to 'no clear change'
'this' approach
'and' maximum duration
– weird sentence – 'even better' is an odd choice of words we changed the wording
comma before the word but
'to represent' IAV
'to represent'
' are not expected'
do you mean 'for' the maximum duration?
'which is probably the most difficult'
what is a 'outlier member'? we added a short description
'with' not after

Section 3.4
difference 'in'
'unstable'
'only possible'

Discussion
Why is the discussion full of really short paragraphs? This reads as a list and could be made flow better and clearer. We discuss several aspects of our work that are not necessarily connected, but need to be addressed to set our results into context. A 'flow' can be nice to have, but we do not think it is necessary. We find the discussion clear and good to read.

– weird sentence – I would say the number of SMILEs available we changed the sentence
– by biases do you mean different mean states? Yes, we added a 'mean'
' Better agreement' - remove the A
– what do you mean by 'in accordance' ? the variability increases in accordance to the increases in the mean; a higher variability can be expected when the mean increases, just because of higher numbers of the respective variable. We changed the wording to 'in conjunction with'
404-406 and 416-419 are difficult to understand We added some information to make it clearer
might be? Or is? Well it is not sure, because we did not specifically test this, but we can also not exclude these uncertainties. We replaced 'might' by 'can'
what does this tell us?
'challenges'

Conclusions
This sentence is hard to read We do not agree; the sentence is two lines long and has a clear meaning and structure
434/449 This was true, however there a now quite a few studies using multiple SMILEs we removed the first sentence (434) and changed the wording in the second sentence to make clear that there is an increasing number of publications comparing SMILEs, although most publications about SMILEs indeed just use one SMILE
remove (dis)

Figure captions:
F4 'shading' and replace 'via' with 'using'
F5 ' during the period'

F5 – mention that individual lines are individual ensemble members I think it is clear enough. The caption says 'and each ensemble member'

F7 – tell us that the reference period is 1980-2009

F8 – is this variance calcualted as square of the std? Yes, this is the definition of variance and standard deviation

F8 how do you define a significant difference? We added the information that this is tested with a Brown-Forsythe Test

F10 how do you pool IAV? The pooled IAV is just a combination of residuals from all members, e.g. 50x30years for CRCM